# The catalytic activity of TCPTP is auto-regulated by its intrinsically disordered tail and activated by Integrin alpha-1

Jai Prakash Singh [1,2,3,8], Yang Li[4,8], Yi-Yun Chen [1,5], Shang-Te Danny Hsu [1,2,6], Rebecca Page[7], Wolfgang Peti [4✉] & Tzu-Ching Meng [1,2,6✉]

T-Cell Protein Tyrosine Phosphatase (TCPTP, PTPN2) is a non-receptor type protein tyrosine phosphatase that is ubiquitously expressed in human cells. TCPTP is a critical component of a variety of key signaling pathways that are directly associated with the formation of cancer and inflammation. Thus, understanding the molecular mechanism of TCPTP activation and regulation is essential for the development of TCPTP therapeutics. Under basal conditions, TCPTP is largely inactive, although how this is achieved is poorly understood. By combining biomolecular nuclear magnetic resonance spectroscopy, small-angle X-ray scattering, and chemical cross-linking coupled with mass spectrometry, we show that the C-terminal intrinsically disordered tail of TCPTP functions as an intramolecular autoinhibitory element that controls the TCPTP catalytic activity. Activation of TCPTP is achieved by cellular competition, *i.e.*, the intrinsically disordered cytosolic tail of Integrin-α1 displaces the TCPTP autoinhibitory tail, allowing for the full activation of TCPTP. This work not only defines the mechanism by which TCPTP is regulated but also reveals that the intrinsically disordered tails of two of the most closely related PTPs (PTP1B and TCPTP) autoregulate the activity of their cognate PTPs via completely different mechanisms.

[1] Institute of Biological Chemistry, Academia Sinica, 128 Academia Road Sec. 2, Nankang, Taipei 115, Taiwan. [2] Chemical Biology and Molecular Biophysics, Taiwan International Graduate Program, Academia Sinica, 128 Academia Road Sec. 2, Nankang, Taipei 115, Taiwan. [3] Department of Chemistry, National Tsing-Hua University, 101 Kuang-Fu Road Sec. 2, Hsinchu 300, Taiwan. [4] Department of Molecular Biology and Biophysics, University of Connecticut Health Center, Farmington, CT 06030, USA. [5] Academia Sinica Common Mass Spectrometry Facilities for Proteomics and Protein Modification Analysis, 128 Academia Road Sec. 2, Nankang, Taipei 115, Taiwan. [6] Institute of Biochemical Sciences, National Taiwan University, 1 Roosevelt Road Sec. 4, Taipei 106, Taiwan. [7] Department of Cell Biology, University of Connecticut Health Center, Farmington, CT 06030, USA. [8] These authors contributed equally: Jai Prakash Singh, Yang Li. ✉email: peti@uchc.edu; tcmeng@gate.sinica.edu.tw

cell protein tyrosine phosphatase (TCPTP, PTPN2) is a ubiquitously expressed protein tyrosine phosphatase (PTP) that was originally cloned from a T cell cDNA library[1]. In humans, TCPTP exists as two isoforms, i.e., TC45 and TC48[2]. Both are composed of an N-terminal PTP catalytic domain (residues 1–300) and a C-terminal tail that differs in length due to alternative mRNA splicing[2,3]. The key difference between TC45 and TC48 is the presence of a ~3 kDa hydrophobic C-terminus in the latter that functions to specifically recruit TC48 to the endoplasmic reticulum[4]. TC45, which lacks the hydrophobic C-terminus, is found in the nucleus and cytoplasm[4,5] (its C-terminus contains a specific nuclear localization sequence[4]). Thus, compared to TC48, TC45 is known to dephosphorylate a broad array of substrates[6].

A key family of TCPTP (hereafter referring to the TC45 isoform throughout the manuscript; Fig. 1a) substrates are receptor tyrosine kinases (RTKs), including the Insulin receptor (IR)[7,8], Epidermal growth factor receptor (EGFR)[9], Platelet-derived growth factor receptor (PDGFR)[10], Vascular endothelial growth factor receptor (VEGFR)[11], Fibroblast growth factor receptor (FGFR3)[12], Colony-stimulating factor-1 receptor (CSF1R)[13], and Hepatocyte growth factor receptor (C-Met)[14]. TCPTP dephosphorylates the RTK activation loop phosphorylation sites, thereby directly controlling RTK activity. Except for IRTK, all other RTKs are established onco-drivers. Consistent with this, for many of these RTKs, drugs have been developed to inhibit unregulated kinase activity[15]. In addition to controlling specific signaling pathways via its ability to modulate RTK activity, TCPTP also regulates distinct signaling pathways by acting on downstream kinases and substrates, including the Janus-activated kinases JAK-1 and JAK-3[16], and their substrates, including signal transducer and activator of transcription STAT-1[17], STAT-3[18], and STAT-6[19]. Clearly, modulating TCPTP activity is a promising strategy for therapeutic intervention in cancer management. Furthermore, beyond RTK signaling pathways, TCPTP also modulates the signaling pathways defined by c-Src[20], LCK[21], Fyn[21], p52[Shc,9], Caveolin-1[22], and TGF-beta receptor type-2[23], i.e., ones associated with malignancies and other disease pathways including diabetes[24], rheumatoid arthritis[25] and inflammatory bowel diseases[26]. Taken together, these data demonstrate that TCPTP is a key regulator of diverse cellular signaling pathways. However, and in striking contrast to its well-established biological importance, little is known regarding the molecular activation and regulation of TCPTP.

Within the PTP family, TCPTP is most closely related to PTP1B (72% identity within the catalytic domain; residues 1–300). We and others have shown that the intrinsically disordered C-terminal tail of PTP1B is critical not only for cellular localization (ER recruitment) but also directly influence PTP1B allostery and thus its activity[27,28]. Indeed, a novel PTP1B inhibitor (MSI-1436; Trodusquemine) modulates PTP1B activity by directly binding the PTP1B C-terminal tail[27]. Previous studies suggested that the TCPTP C-terminal tail may also influence the activity of TCPTP[29]. Namely, the activity of TC45 was reported to become fully activated in response to limited proteolysis[29], suggesting that the TCPTP C-terminal tail may have a currently unexplored autoinhibitory function. Moreover, collagen-binding integrin α1 (ITGA1) has been shown to activate full-length TCPTP via the cytoplasmic tail of ITGA1[22,23,30]; however, the mechanism by which this activation is achieved is also unknown. Together, these data demonstrate that a comprehensive, molecular understanding of the potential inhibitory/regulatory function of the TCPTP C-terminal tail may provide a powerful novel route for modulating TCPTP-specific signaling pathways and, in turn, new treatments for diseases like cancer and diabetes.

Here we used an array of orthogonal molecular biology, biochemistry and molecular techniques, in particular biomolecular

NMR spectroscopy, to determine the modes of action of TCPTP autoinhibition and activation. Our data show that TCPTP is regulated by an intrinsically disordered auto-inhibitory domain and explain how this inhibition is achieved at a molecular level. Furthermore, we also show that the ITGA1-mediated activation of TCPTP activity is achieved by the ability of the ITGA1 cyto-plasmic tail to displace the TCPTP autoinhibitory domain. Together, these studies provide key insights into the molecular basis of TCPTP activation and provide a novel route for the development of therapeutics that can activate TCPTP activity via autoinhibitory domain displacement.

## Results

**The TCPTP C-terminal tail is intrinsically disordered**. We used biomolecular NMR spectroscopy to determine the molecular basis for TCPTP autoinhibition. The 2D [$^1$H,$^{15}$N] TROSY spectra of $TCPTP_{CAT}$ (TCPTP catalytic domain residues 1–302; Fig. 1a, b) is of high quality, which allowed the sequence-specific back-bone assignment to be readily completed. The chemical shift index (CSI), calculated from the deviations of Cα and Cβ chemical shifts from random coil values (RefDB[31]), correlates well with secondary structures observed in the crystal structure of $TCPTP_{CAT}$ (Supplementary Fig. 1). We then compared the 2D [$^1$H,$^{15}$N] TROSY spectra of $TCPTP_{CAT}$ and TCPTP (Fig. 1b; full-length TCPTP residues 1–387; Fig. 1a). The $H^N$/N cross-peaks of $TCPTP_{CAT}$ overlap well with their corresponding peaks in the TCPTP spectrum, demonstrating that the conformation of $TCPTP_{CAT}$ is identical in both constructs. Further, the spectra showed additional peaks in the TCPTP spectrum that clustered in the center of the spectrum between 7.5 and 8.5 ppm ($^1$H dimension). This is typical for flexible, mostly unstructured amino acids, demonstrating that the C-terminal tail of TCPTP is an intrinsically disordered region (IDR) and thus mirrors the behavior of the C-terminal tail of PTP1B[27]. Notably, the secondary structure propensity (SSP) scores derived from the chemical shift index (CSI) show that the $TCPTP_{Tail}$ (residues 303–387; Fig. 1a) are not entirely unstructured, but rather contain two partially populated α-helices comprised of residues 343–354 (α8′; ~50% populated), and 374 – 378 (α9′; ~50% populated) (Fig. 1c). The SSP values for these residues are identical in both TCPTP and $TCPTP_{Tail}$ constructs, demonstrating that these propensities are independent of $TCPTP_{CAT}$. The partially populated secondary structural elements in TCPTP differ from those identified in PTP1B, highlighting that both the sequence and the ensemble of structures are different between the two closely related PTPs[27].

**The TCPTP C-terminal tail adopts an ensemble of structures in solution**. To gain further insights into the solution structure of TCPTP, we used liquid chromatography coupled with small-angle x-ray scattering (LC-SAXS). SAXS data for $TCPTP_{CAT}$ (residues 1–314 for SAXS data collection) reported a radius of gyration ($R_g$) of 27.2 ± 0.1 Å, consistent with a ~38 kDa globular protein (Supplementary Fig. 2a, c and Supplementary Table 1). In contrast, the $R_g$ value for TCPTP was 37.3 ± 0.1 Å, ~10 Å larger than that measured for $TCPTP_{CAT}$ and larger than that expected for a mostly globular protein of ~45 kDa. Thus, these SAXS data are in full agreement with the NMR data and confirm that the TCPTP C-terminal IDR region is extended and dynamic in solution (Supplementary Fig. 2a, b and Supplementary Table 1). To gain further insights into the ensemble of structures potentially adopted by TCPTP in solution, we used multi-state modeling (MultiFoXS[32]). MultiFoXS allows one or more conformations of a model to be used to identify the best fit to an experimental SAXS profile. Our data show that using of an ensemble of structures

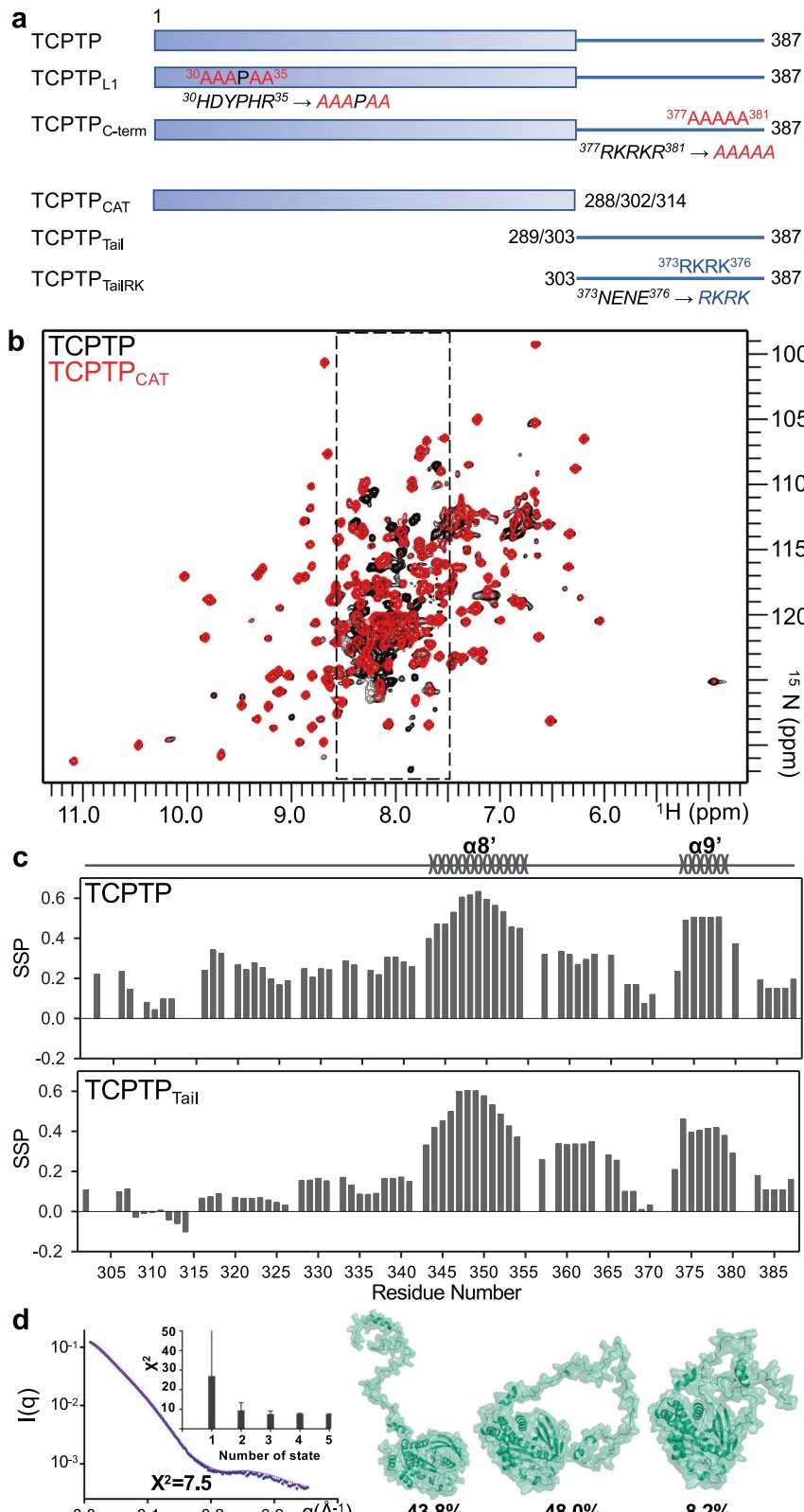

**Fig. 1 TCPTP is comprised of a structured N-terminal catalytic domain and an intrinsically disordered dynamic C-terminal domain. a** TCPTP construct. **b** Overlay of the 2D [$^1$H,$^{15}$N] TROSY spectra of ($^2$H,$^{15}$N)-labeled TCPTP (black) and TCPTP$_{CAT}$ (red). TCPTP IDR peaks (residues 303–387) are highlighted by dashed box. **c** Secondary-structure propensity (SSP) vs TCPTP residue in TCPTP and the isolated TCPTP$_{Tail}$. SSP reports transient secondary structure elements (SSP > 0, α helix; SSP < 0, β strand). **d** TCPTP structural models (right) generated by MultiFoXS[32] based on SAXS data (left). SAXS scattering data (blue) is best represented by a 3-model ensemble (light pink). Inset bar graph highlighting that a 3-member ensemble is statistically necessary for the best fit ($X^2$). The error bar indicates the range of $X^2$ values for top 100 multistate models[32]. The computed output value along with error from MultiFoXS is provided in the Source Data file as Source data. The three ensemble members are shown in green; populations for the optimal fit are shown.

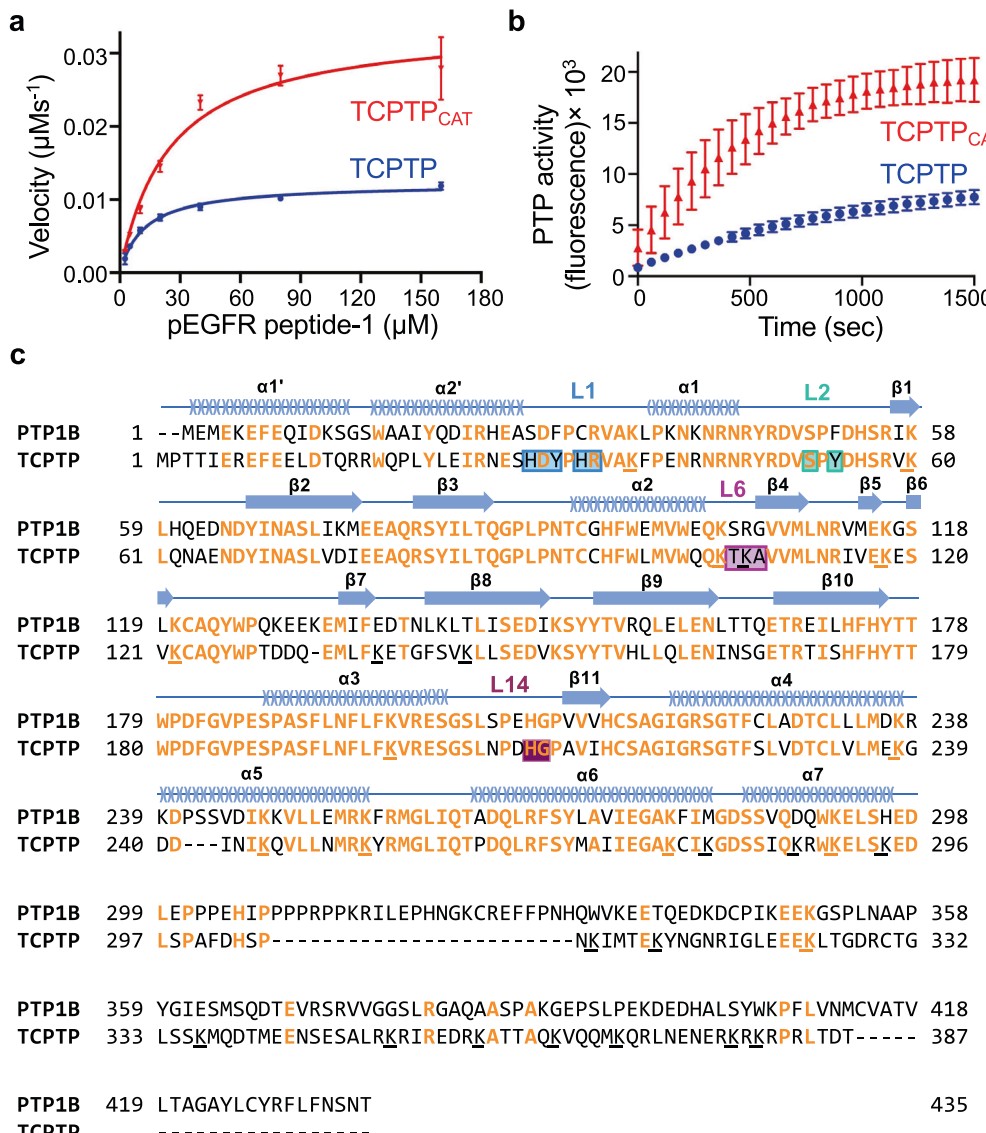

**Fig. 2 The TCPTP C-terminal tail autoinhibits TCPTP. a** Comparison of the catalytic activity between TCPTP with TCPTP$_{CAT}$ (residues 1–314) vs pEGFR peptide-1, derived from the EGFR cytoplasmic tail. **b** Comparison of the catalytic activity between TCPTP with TCPTP$_{CAT}$ (residues 1–314) vs DiFMUP. Data in (**a**) and (**b**) are presented as the mean of nine independent reactions ($n = 9$; mean ± SE). **c** Sequence alignment of TCPTP and PTP1B. Identical residues are highlighted in orange, lysines are underlined; residues exhibiting CSP due to the TCPTP$_{Tail}$ are boxed with corresponding loops indicated (colored as in Figs. 3 and 4); secondary structural elements are shown above the sequence. Source data are provided as a Source Data file.

significantly improves the fit between calculated and experimental data, with a three-state model showing the statistically most meaningful minimal ensemble, which includes one highly extended confirmation (43.8%) and two more compact conformations (48% and 8.2%) (Fig. 1d). Together, the NMR and SAXS data demonstrate that the TCPTP C-terminal tail is highly dynamic in solution and adopts an ensemble of interchanging structures.

**The C-terminal tail of TCPTP negatively regulates TCPTP catalytic activity.** We next investigated the role of the TCPTP C-terminal dynamic tail on TCPTP catalytic activity by comparing the activities of TCPTP and TCPTP$_{CAT}$. The catalytic activity was tested using phosphopeptides derived from either the cytoplasmic tail or the activation loop of the EGF receptor, a known substrate of TCPTP[9]. For both substrate peptides, the rate of dephosphorylation by TCPTP was ~3-fold slower than that measured for TCPTP$_{CAT}$ (Fig. 2a and Supplementary Fig. 3a).

Michaelis-Menten analysis revealed that the C-terminal tail of TCPTP affects both the $k_{cat}$ and K$_M$ (Table 1 and Supplementary Table 2). These data are consistent with an autoinhibitory function of the TCPTP C-terminal tail and suggest that the change in activity is not achieved by completely blocking the TCPTP active site, but instead via a yet unknown mechanism.

To further test the extent of TCPTP active site engagement/blockage, we repeated the experiment with the small molecule 6,8-Difluoro-4-Methylumbelliferyl Phosphate (DiFMUP). The data mirrored those obtained for the EGFR peptide substrates (Fig. 2b). Namely, DiFMUP dephosphorylation by TCPTP was also ~3-times slower than with TCPTP$_{CAT}$. These data confirmed that the TCPTP C-terminal tail influences the activity of TCPTP, without directly binding the active site. Interestingly, the autoregulatory role of the TCPTP C-terminal tail is unique to TCPTP, as this behavior of the C-terminal tail is not observed in its most closely related PTP, PTP1B (Supplementary Fig. 3b). Consistent with this observation, a sequence alignment between

**Table 1 Enzymatic activity assay (Michaelis-Menten analysis).**

| | $k_{cat}$ (s$^{-1}$) | $K_m$ (μM) | $k_{cat}/K_m$ (μM$^{-1}$s$^{-1}$) | N |
|---|---|---|---|---|
| *TCPTP* | | | | |
| TCPTP$_{CAT}$ | 34.0 ± 0.9 | 24.8 ± 2.0 | 1.4 ± 0.1 | 9 |
| TCPTP$_{C-term}$ | 23.7 ± 0.5 | 17.9 ± 1.1 | 1.3 ± 0.1 | 12 |
| TCPTP$_{L1}$ | 19.3 ± 0.8 | 21.0 ± 2.7 | 0.9 ± 0.1 | 12 |
| TCPTP | 12.2 ± 0.2 | 12.4 ± 0.6 | 1.0 ± 0.1 | 9 |
| *ITGA1 (saturated)* | | | | |
| TCPTP$_{CAT}$ + ITGA1_TR | 31.1 ± 0.7 | 23.0 ± 1.6 | 1.3 ± 0.1 | 9 |
| TCPTP + ITGA1_FCT | 26.6 ± 0.8 | 20.2 ± 1.9 | 1.3 ± 0.1 | 9 |
| TCPTP + ITGA1_TR | 26.2 ± 0.6 | 15.9 ± 1.3 | 1.6 ± 0.1 | 9 |

pEGFR peptide-1, derived from the cytoplasmic tail of EGFR, was used as a substrate. Calculated kinetic parameters are presented as the mean of nine or twelve independent reactions ($n = 9$ or $n = 12$; mean ± standard error, SE; explicit n number for individual analysis is shown in the column under the header 'N'). Source data are provided as a Source Data file.

both enzymes shows that while TCPTP and PTP1B share high sequence identity in their catalytic domains (72% identity), their C-terminal tails differ significantly (Fig. 2c; 14% identity). Thus, the distinct intrinsically disordered tails of TCPTP and PTP1B permit each enzyme to be differentially activated and regulated.

**Residues 344–385 constitute the TCPTP autoinhibitory domain.** An in trans interaction of the IDR C-terminal tail of TCPTP leads to a dose-dependent inhibition of the TCPTP catalytic activity[29]. Thus, we tested if the TCPTP$_{Tail}$ peptide interacts directly, in trans, with TCPTP$_{CAT}$. The addition of increasing amounts of TCPTP$_{CAT}$ (1:0 to 1:20, TCPTP$_{TAIL}$: TCPTP$_{CAT}$) resulted in reduced intensities of multiple peaks, corresponding to residues 344–385, in the 2D [$^{1}$H,$^{15}$N] HSQC spectrum of TCPTP$_{Tail}$ (Supplementary Fig. 4). TCPTP residues 350–385 are enriched in positively charged residues and thus we hypothesized that they may facilitate TCPTP$_{CAT}$ binding via one or more dynamic protein: protein (fuzzy) interactions[33], similar to those used by the Ser/Thr protein phosphatases PP2A[34] and PP2B[35], among others (Fig. 2c). In PP2A/B, IDR charged residues interact dynamically with patches of residues of the opposite charge present in folded and/or IDP cognate binding partners and allow for substrate binding and catalytic activity regulation. To test if these positively charged TCPTP residues play a role in binding, we generated a TCPTP$_{Tail}$ construct with an increased pI (more positive charge), which we achieved by mutating residues $^{373}$NENE$^{376}$ to $^{373}$RKRK$^{376}$ (TCPTP$_{TailRK}$). The CSI analysis showed that TCPTP$_{TailRK}$ has a secondary structure profile identical to TCPTP$_{Tail}$ (Fig. 3a), demonstrating no large differences in the preferred secondary structure. We then repeated the NMR titration experiment and observed a reduction in the peak intensities in the 2D [$^{1}$H,$^{15}$N] HSQC spectrum of TCPTP$_{TailRK}$ at a lower ratio (1:10) than with TCPTP$_{Tail}$, consistent with a tighter interaction (Fig. 3b, c). These data highlight the importance of charged residues in the interaction of the TCPTP C-terminal tail and the catalytic domain.

**The TCPTP autoinhibitory domain binds TCPTP$_{CAT}$ via two surfaces that are distal from the active site.** To determine the complementary binding site of TCPTP tail residues 344–385 on TCPTP$_{CAT}$, we performed reverse titration experiments. Namely, we added unlabeled TCPTP$_{TailRK}$ to ($^{2}$H,$^{15}$N)-labeled TCPTP$_{CAT}$ and recorded 2D [$^{1}$H,$^{15}$N] TROSY spectra. Small, but clear chemical shift perturbations (CSPs) were readily identified (Fig. 3d). The CSPs map to 4 loops in TCPTP, which define two distinct surfaces: (1) the N-surface, defined by loops L1 and L2,

which is adjacent to the TCPTP active site and includes part of the N-terminal domain that is specific to PTP1B and TCPTP and (2) the back or B-surface, defined by loops L6 and L14, which is on the backside of TCPTP and with L14 loop separated from the catalytic PTP loop by β-strand β11 (Fig. 3e). Two residues in β-strand β8, E148 and D149, also exhibited CSPs. None of these secondary structural elements has been previously identified to have a role in PTP catalysis and/or substrate recognition, generally, nor in PTP1B/TCPTP, specifically. Finally, the observed CSPs are fully consistent with the activity assay data; namely, the autoregulatory domain binds outside the TCPTP active site.

To confirm this interaction also occurs in cis, we used chemical cross-linking coupled with mass spectrometry (CX-MS) using full-length TCPTP[36]. To achieve comprehensive coverage, we used two distinct cross-linkers, disuccinimidyl sulfoxide (DSSO) and bissulfosuccinimidyl suberate (BS3), in independent experiments; both cross-linkers have spacer arms ~11 Å. We mixed purified full-length TCPTP with increasing concentrations of cross-linker. As the concentrations of both cross-linker were increased, the bands corresponding to TCPTP shifted towards ones with increased mobility (Fig. 4a). To identify the cross-linked lysine residues, the cross-linked TCPTP bands were excised from the SDS-PAGE gel and subjected to MS analysis. A map of the peptides cross-linked between TCPTP$_{CAT}$ and the TCPTP C-terminal tail residues is shown in Fig. 4b. Of the 20 lysine residues present in TCPTP$_{CAT}$, 4 (K38, K60, K107, and K118) were identified to be cross-linked with multiple lysines from the TCPTP C-terminal tail (Fig. 4b and Supplementary Table 3). Critically, three of these lysines correspond to lysines either immediately adjacent (K38), between (K60) or part of (K107) the N- and B-surfaces, respectively demonstrating that these data are fully consistent with the NMR-based interaction analysis. In addition to K38, K60 and K107, the CX-MS study also showed that K118, a residue in the TCPTP E-loop, is also sufficiently close the TCPTP C-terminal tail to result in cross-linking. The E-loop is adjacent to the active site and has been indicated to play a role in substrate recruitment for different PTPs[37]. Like the residues that define the N-surface, K118 is adjacent to the TCPTP active site. Finally, the lysine residues in the C-terminal tail observed to be cross-linked most frequently to lysines in TCPTP$_{CAT}$ correspond to K358, K364 and K369, i.e., the same residues determined to bind TCPTP$_{CAT}$ using NMR spectroscopy (residues 344–385). Together, the cross-linking and NMR-based interaction studies show that TCPTP dynamic C-terminal IDR residues 344–385 interact directly with the TCPTP$_{CAT}$ domain.

As a final control, we repeated the cross-linking experiment between the autoinhibitory and catalytic domains in trans using the individual TCPTP$_{CAT}$ and TCPTP$_{Tail}$ domains, i.e. in an experimental manner identical to that used for the NMR experiments. As expected, the results were fully consistent with both the NMR spectroscopy and the in cis cross-linking data. Namely, as observed for full-length TCPTP, TCPTP$_{CAT}$ residues K38, K107, and K118 were again cross-linked to multiple lysines in the TCPTP$_{Tail}$ peptides, with the greatest number of cross-links observed for TCPTP$_{Tail}$ residues K358, K364, and K369 (cross-links were also observed for K290 and K294, but this is likely due to the increased accessibility of these residues in the free IDP [TCPTP$_{Tail}$] versus the same residues within the context of the full-length protein) (Supplementary Fig. 5a, b and Supplementary Table 4). Together, the cross-linking and NMR-based interaction studies show that TCPTP dynamic C-terminal IDR residues 344–385 interact directly with the TCPTP$_{CAT}$ domain, with the TCPTP C-terminal tail 344–385 wrapping around the outer edge of the catalytic domain to engage the N- and B-surfaces in a highly specific manner.

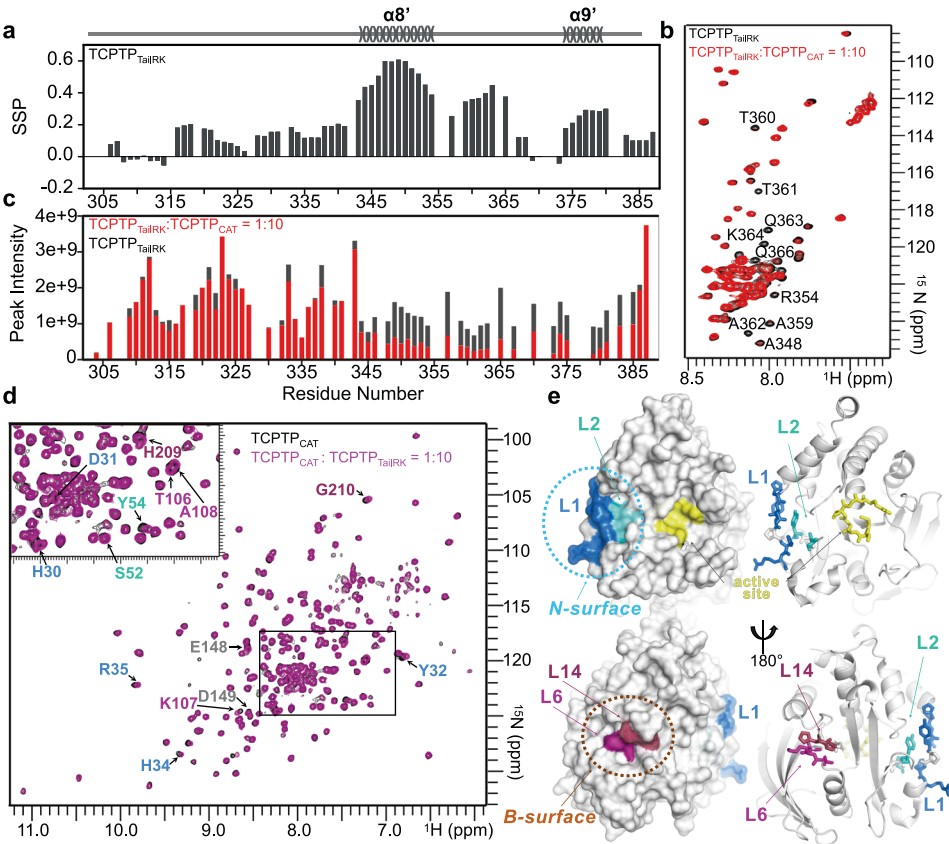

**Fig. 3 The TCPTP dynamic C-terminal tail binds the TCPTP catalytic domain. a** SSP vs TCPTP$_{TailRK}$ residues. SSP highlights transient secondary structure elements in the TCPTP$_{TailRK}$ C-terminal tail (SSP > 0, α helix; SSP < 0, β strand). **b** Overlay of the 2D [$^1$H,$^{15}$N] HQSC spectra of $^{15}$N-labeled TCPTP$_{TailRK}$ (black) and TCPTP$_{CAT}$-bound TCPTP$_{TailRK}$ (red). Residues with significant reductions in cross peak intensity are annotated. **c** Peak intensity comparison between free TCPTP$_{TailRK}$ (black) and TCPTP$_{CAT}$-bound TCPTP$_{TailRK}$ (red). **d** Overlay of the 2D [$^1$H,$^{15}$N] TROSY spectra of ($^2$H,$^{15}$N)-labeled TCPTP$_{CAT}$ (black) and TCPTP$_{TailRK}$-bound TCPTP$_{CAT}$ (magenta). Peaks with chemical shift perturbations (CSPs) are labeled (colors corresponding to residues in L1, L2, L6, L14 as in **e**). **e** Residues with CSPs in **d** are mapped onto the 3D structure of TCPTP$_{CAT}$ (PDB: 1L8K). Residues cluster in two regions, the N-surface (loop L1, blue; L2, cyan) which is to the left edge of the active site (yellow) and the B-surface (L6, magenta; L14, raspberry), which is on the opposite face as the active site.

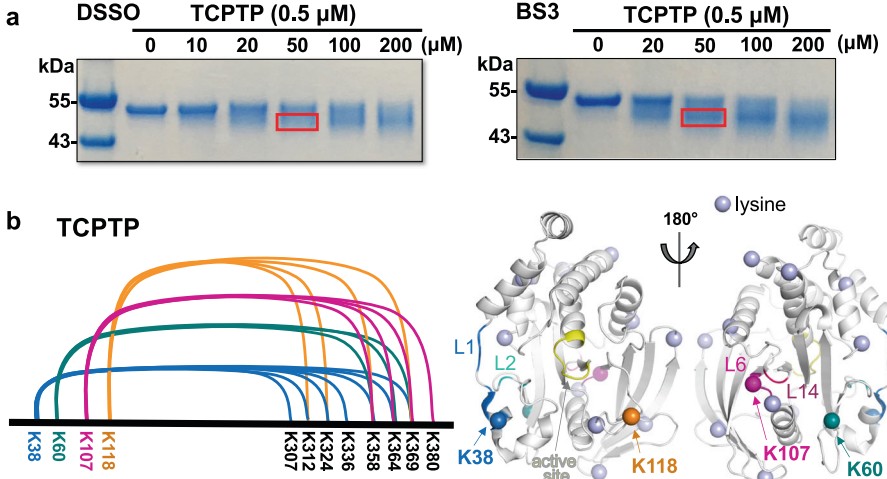

**Fig. 4 TCPTP CX-MS demonstrates that the dynamic C-terminal tail binds specifically to the TCPTP$_{CAT}$ domain. a** Intramolecular chemical cross-linking of TCPTP by DSSO (*Left*; SDS-PAGE) and BS3 (*Right*; SDS-PAGE). Each lane on SDS-PAGE is from an independent reaction mixture at the indicated concentration of cross-linker, labeled on top in μM unit. The cross-linked product showed higher mobility compared to the control sample without a cross-linker. The red box indicates excised cross-linked product from SDS-PAGE for MS analysis. **b** Cross-linking map of the TCPTP lysine residues intramolecularly cross-linked between the TCPTP catalytic domain and the dynamic C-terminal tail. Cross-linked lysine residues (K38, K60, K107, K118) are mapped on 3D structure of TCPTP$_{CAT}$ (PDB ID: 1L8K). All lysine residues present in TCPTP$_{CAT}$ are shown as lavender spheres. Source data are provided as a Source Data file.

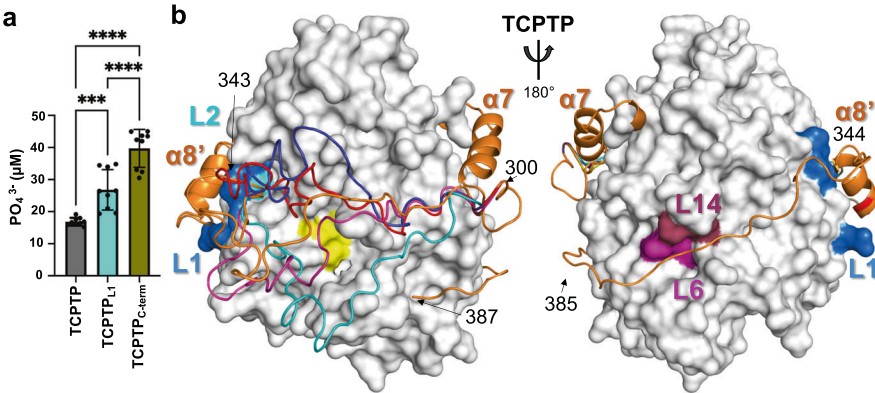

**Fig. 5 Structural model of TCPTP autoinhibition. a** Bar graph reporting changes in TCPTP catalytic activity due to mutations in the catalytic domain (TCPTP_L1) and the C-terminal tail (TCPTP_C-term) using pEGFR peptide-1 as a substrate derived from the EGFR cytoplasmic tail. Data are presented as the mean of nine independent reactions ($n = 9$; mean ± standard deviation, SD). *** indicates *p*-value of <0.001 ($p = 0.0009$) and **** indicates *p*-value of <0.0001, obtained from one-way ANOVA analysis with Tukey's multiple comparison test. **b** Model of TCPTP autoinhibition that integrates the NMR, CX-MS and activity data. Residues ~300–340 from the TCPTP C-terminal tail adopt an ensemble of conformations (depicted in different colors: cyan, navy, magenta, red, and yellow), akin to a windshield wiper, to dynamically block substrate access to the TCPTP active site (yellow). Residues with CSPs are colored as in Fig. 3e and show how residues 344–385 bind the backside of TCPTP. Source data are provided as a Source Data file.

**TCPTP autoinhibition**. NMR spectroscopy and CX-MS established that TCPTP C-terminal residues 344–385 interact directly with TCPTP loops L1, L2, L6, L14, and β8. To determine if disrupting these interactions, either on the catalytic or C-terminal domain, negatively impacts TCPTP autoinhibition, we used mutagenesis coupled with activity assays. Loop L1 is part of the N-terminal domain of TCPTP that is exclusive to TCPTP and PTP1B and so far has no reported biological function. To test the importance of L1 for TCPTP C-terminus binding and autoinhibition, we generated a TCPTP variant in which L1 residues were mutated to alanine (TCPTP_L1, H30A/D31A/Y32A/H34A/R35A, Fig. 1a). As expected, compared to TCPTP, TCPTP_L1 showed increased catalytic activity, i.e., reduced autoinhibition (Fig. 5a). Similarly, we also generated a TCPTP C-terminal mutant (TCPTP_C-term), in which we mutated a positively charged patch within TCPTP residues 344–385, 377RKRKR381 to 377AAAAA381 (Fig. 1a). Based on our NMR data, we predicted that the loss of these positively charged amino acids would weaken the interaction between the TCPTP C-terminus and TCPTP_CAT and, in turn, result in a decrease in TCPTP autoinhibition. As predicted, this variant showed an increase in TCPTP activity (Fig. 5a). Next, we created variants to highlight the importance of charged residues in loops L1 and L6 for binding with the TCPTP C-terminal tail (Supplementary Fig. 6a, b).

Together with the molecular data derived from the NMR spectroscopy and CX-MS experiments, these data allow us to propose a model for the binding of the TCPTP C-terminal tail to TCPTP_CAT. Our NMR data show that TCPTP residues 344–385 bind to the N- and B-surfaces on TCPTP_CAT. In addition to confirming these interactions, the CX-MS data also identified a transient interaction with the E-loop residue K118, a residue near those in β8 that also exhibited CSPs by NMR spectroscopy. Thus, this leads to a model in which TCPTP residues 300–343 form a flexible linker across the front of the active site between TCPTP helix α7 and TCPTP residues 344–385 (Fig. 5b). Because this linker is flexible, it readily interchanges between an ensemble of conformations, functioning akin to a dynamic 'windshield wiper' that reduces substrate access to the TCPTP active site without blocking it completely. This mechanism of dynamic inhibition was also recently seen to occur in the ser/thr protein phosphatase PP2B (calcineurin) when bound to its regulators and substrates NHE1[35] and RCAN1[38] suggesting that dynamically occluding substrate access to the active site by tethered substrates and/or inhibitors may be a general mode of (auto)inhibition used by enzymes.

**ITGA1 activates TCPTP by competing with the TCPTP C-terminal tail**. After discovering the mechanism of TCPTP's autoinhibition by its intrinsically disordered C-terminus, we set out to identify how TCPTP is activated. Integrin α-1 (ITGA1) has been reported to activate TCPTP and to enhance TCPTP-mediated substrate dephosphorylation[22,23,30]. To determine how ITGA1 activates TCPTP, we first incubated TCPTP with peptides derived from the ITGA1 cytoplasmic tail and measured TCPTP activity. Like the TCPTP C-terminal tail, the ITGA1 cytoplasmic tail is an IDR enriched in positively charged residues. We generated three peptides that differ in their length and charge: ITGA1_FCT (ITGA1 full cytoplasmic tail), ITGA1_TCT (ITGA1 truncated cytoplasmic tail; 55% positively charged residues) and ITGA1_TR (ITGA1 tandem repeat; 2x ITGA1_TCT; containing two patches of positively charged residue). EC_50 assays showed the strongest activation by ITGA1_TR, followed by ITGA1_FCT and ITGA1_TCT (Fig. 6a). This suggests that, as observed for the TCPTP C-terminal tail, positive charge is critical for ITGA1-mediated stimulation of TCPTP activity. Further, since ITGA1_FCT functions better than ITGA1_TCT, these data also show that residues beyond the charged region contribute to the interaction.

To determine how ITGA1 binds TCPTP, we titrated the ITGA1_FCT and ITGA1_TR peptides into ($^2$H,$^{15}$N)-labeled TCPTP_CAT and recorded 2D [$^1$H,$^{15}$N] TROSY spectra. Both titrations revealed significant CSPs in the same residues that exhibit shifts when TCPTP_TailRK titrations (Fig. 6b and Supplementary Fig. 7a). This demonstrates that ITGA1 and the TCPTP_TailRK bind TCPTP_CAT in a similar manner. Further, the ITGA1 peptide titrations result in larger CSPs and those with the TCPTP_TailRK, suggesting the ITGA1 interactions are stronger and thus allowing for the displacement of the TCPTP C-terminal tail (Fig. 6c).

To determine if the displacement of TCPTP_Tail leads to activation, we repeated the activity measurements of TCPTP and TCPTP_CAT in the presence of ITGA1 peptides using the two phosphopeptides synthesized from EGFR and DiFMUP as substrates. Both TCPTP and TCPTP_CAT were saturated with excess ITGA1 peptide (Fig. 6d and Supplementary Fig. 7b) and

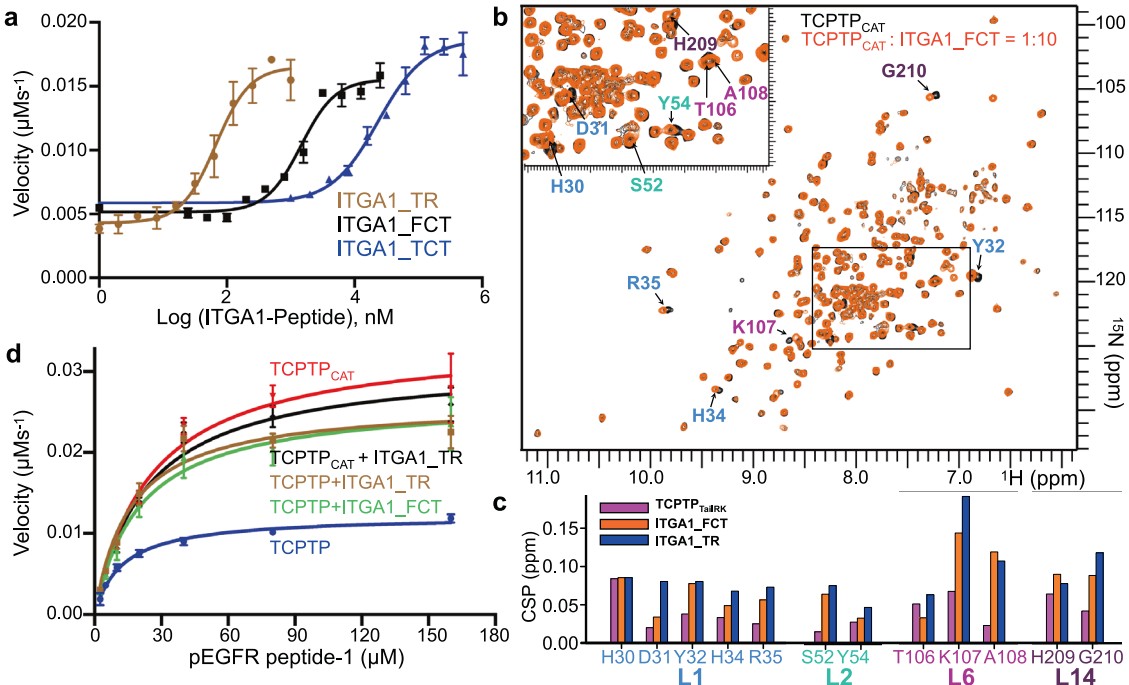

**Fig. 6 Integrin α1 (ITGA1) activates TCPTP by displacing the TCPTP autoinhibitory domain. a** EC$_{50}$ assays of ITGA1 peptides vs TCPTP, monitored by dephosphorylation of pEGFR peptide-1 as substrate derived from the EGFR cytoplasmic tail. Data are presented as the mean of three independent reactions ($n = 3$; mean ± SE). **b** Overlay of the 2D [$^1$H,$^{15}$N] TROSY spectra of ($^2$H,$^{15}$N)-labeled TCPTP$_{CAT}$ (black) and in complex with ITGA1_FCT (orange). Residues with CSPs are annotated. **c** CSP comparison for residues in TCPTP$_{CAT}$ loops L1, L2, L6 and L14 (colored as in Fig. 3e) when in complex with either TCPTP$_{TailRK}$, ITGA1_FCT or ITGA1_TR peptides. **d** Catalytic activity of TCPTP and TCPTP$_{CAT}$ (residues 1–314) in the presence or absence of ITGA1 peptides (ITGA1_TR or ITGA1_FCT); same substrate as described in **a**. Data are presented as the mean of nine independent reactions ($n = 9$; mean ± SE). Source data are provided as a Source Data file.

the dephosphorylation was measured. In the presence of ITGA1 peptide, the rate of TCPTP dephosphorylation increased by ~3-fold, while that of TCPTP$_{CAT}$ was unchanged. Further, the rate of TCPTP dephosphorylation in the presence of ITGA1 peptide was identical to that for TCPTP$_{CAT}$, i.e., the same as that when no autoinhibitory domain is present (Fig. 6d and Supplementary Fig. 7b). These activity results were identical for both peptide substrates and DiFMUP (Supplementary Fig. 7c). Together, these data demonstrate that ITGA1 activates TCPTP by competing off the autoregulatory C-terminal TCPTP tail, allowing the TCPTP active site to become fully accessible and thus achieve maximum activity.

## Discussion

Receptor tyrosine kinases (RTK) are well-established drug targets for a broad variety of human diseases, including cancer[15,39]. Unfortunately, patients commonly develop resistance against RTK inhibitors, rapidly rendering these drugs ineffective[15]. Thus, additional routes to suppress RTK-directed pathways are critically needed. TCPTP is a well-known negative regulator of oncodriver RTKs, including EGFR, PDGFR, VEGFR, cMET, FGFR3, and CSF1R. An alternative approach to suppress RTK-driven onco-genic signaling is to activate TCPTP, which has been shown to have unusually low cellular activity. Remarkably, more than two decades ago, a report suggested that this low cellular activity of TCPTP may be a consequence of autoinhibition[29]. However, to date, no studies confirming TCPTP autoinhibition, much less describing an underlying mechanism of TCPTP's autoinhibition, have been described.

Here, we show that the intrinsically low activity of TCPTP is due to autoinhibition by its dynamic C-terminal IDR tail. Further,

we demonstrate that TCPTP activation by IGTA1 is achieved by displacing the TCPTP IDR C-terminal tail, subsequently rendering the TCPTP active site fully accessible to substrates. This was unexpected as its most similar homolog, PTP1B, is not autoinhibited. TCPTP and PTP1B share a similar architecture that includes a unique N-terminal domain (helices α1′, α2′ and loops L0/L1) with no known function, the core catalytic domain, helix α7 that extents the core catalytic domain and is essential for allostery in PTP1B[28] and a long IDR C-terminal tail[27]. Because of their high sequence similarity, it was assumed these proteins functioned similarly. However, our data shows that the IDR C-terminus in PTP1B and TCPTP have highly distinct functions. In PTP1B, the IDR tail is responsible for recruiting PTP1B to the ER (via an ER-binding helix) and enhancing PTP1B allostery. Indeed, we and others have reported that small molecular com-pounds can bind the PTP1B IDR to directly influence PTP1B allostery and, in turn, activity[27]. In contrast, as we show here, the IDR C-terminal tail in TCPTP functions completely differently. Namely, the TCPTP IDR C-terminal tail interacts directly with the TCPTP catalytic domain to maintain TCPTP in a low activity state. However, this is not achieved by binding and blocking the active site. Rather, TCPTP residues 344–385 bind to the TCPTP catalytic domain on two surfaces distal from the active site, an interaction that is sufficient to keep the flexible portion of the IDR, residues 300–343, in a position that dynamically occludes substrates from the TCPTP active site (Fig. 5b). The dynamic, interchanging conformations of these flexible residues are akin to a 'windshield wiper' in a car, with the dynamic residues moving back and forth across the front of the enzyme to sterically exclude substrates from efficiently reaching the active site. While this does not fully inhibit TCPTP, it significantly slows down the rate of dephosphorylation by blocking substrate access to the catalytic

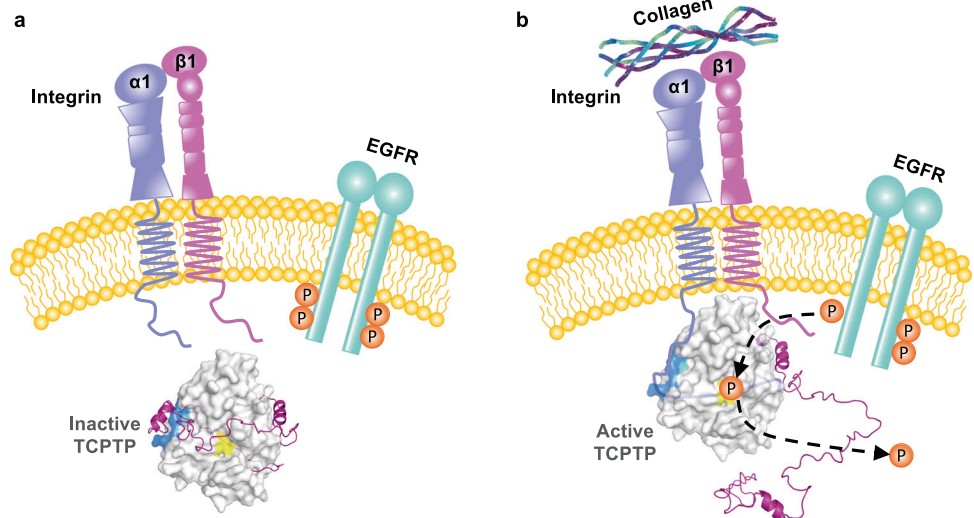

**Fig. 7 TCPTP autoinhibition and activation is driven by intrinsically disordered protein: protein interactions. a** Inactive state of TCPTP - Intramolecular interaction of the IDR TCPTP C-terminal tail with TCPTP$_{CAT}$ via the N-surface (navy) leads to the formation of TCPTP's autoinhibitory conformation. **b** Active form of TCPTP – the IDR C-terminal cytoplasmic tail of ITGA1 binds TCPTP$_{CAT}$ via loops L1, L2 and displaces the intramolecular interaction with the IDR TCPTP C-terminal tail, which relieves the autoinhibitory conformation, leading to efficient dephosphorylation of EGFR. Recruitment and activation of TCPTP by ITGA1 requires collagen binding to integrin α1β1[30], as a result, conformational changes occur in the integrin α1β1 heterodimer, which allow the ITGA1 cytoplasmic tail to become exposed from β1[53], leading to recruitment and activation of TCPTP.

pocket. A similar mode of action was recently described for the ser/thr protein phosphatase PP2B/Calcineurin by its substrate NHE1[35]. These data suggest that this emerging mechanism of dynamic exclusion of active site accessibility likely occurs throughout biology.

So how is TCPTP autoinhibition relieved in a cell? For full cellular activity, TCPTP must be both targeted to its point of action, which are RTKs, and the inhibitory IDR tail must be displaced from the catalytic domain. Here, we show that both are achieved by ITGA1 (Fig. 7). Using NMR spectroscopy and activity assays, we demonstrated that the IDR intercellular tail of ITGA1 efficiently competes with TCPTP residues 344–385 for the identical binding surfaces on the TCPTP catalytic domain. This displacement not only allows for the recruitment of TCPTP to its cellular point of action but also releases TCPTP from its IDR tail-mediated autoinhibition, resulting in full TCPTP activation.

To determine activation of TCPTP via tail displacement is achieved by all or only specific integrins, we aligned the sequences of the cytoplasmic tails of multiple members of the Integrin family to assess their conservation. Interestingly, ITGA11 possesses a cluster of positively charged residues (RSARRRR) similar to ITGA1 (Supplementary Fig. 8a, b). In contrast, the cytoplasmic tail of ITGA10 has a large cluster of negatively charged residues (EEEKREEK). Using peptides derived from ITGA10 and ITGA11 we tested if either of these peptides could activate TCPTP. Indeed, ITGA11, as observed for ITGA1, efficiently activates TCPTP, while ITGA10 fails to do so (Supplementary Fig. 8b). This highlights the importance of positively charged residues for displacing the autoinhibitory TCPTP C-terminal tail. Interestingly, ITGA1 and ITGA11 belong to the subfamily of collagen-binding Integrins. In line with the previous finding that engagement with collagen is a prerequisite for ITGA1-mediated activation of TCPTP[30], our data show that ITGA11 likely also activates TCPTP once it binds to the extracellular matrix (Fig. 7). Clearly, additional experiments are necessary to unravel the exact functional consequence of ITGA1 or ITGA11-mediated activation of TCPTP in cells. While the TCPTP C-terminal tail has the strong enrichment in the positively charged residues needed to achieve autoinhibition, PTP1B, has only a pro-rich C-terminal tail

without a positively charged cluster of residues. Further, the TCPTP catalytic domain residues identified by NMR spectroscopy to bind the C-terminal tail are only moderately conserved between the two PTPs (Fig. 2c). These sequence differences, especially those in the C-terminal tail, explain the distinct functionalities of the IDR tails of these exceptionally similar PTPs.

Finally, this discovery highlights a powerful new strategy for drug development that targets TCPTP specifically. Namely, drugs that displace the IDR C-terminal tail to enhance TCPTP activity in cells. Indeed, a previous high throughput screen in search of novel TCPTP activators reported that five out of six hits of chemical compounds were positively charged[40], consistent with our discovery of the importance of positively charged residues for both TCPTP-mediated autoinhibition and ITGA1-mediated activation. Thus, our data not only reveal a new regulatory role by the TCPTP C-terminal tail, which achieve inhibition via an emerging mechanism of dynamic occlusion but also highlight an exciting avenue for the design of new compounds that either weaken or strengthen TCPTP's autoinhibitory state and can be used to counteract the dysregulation associated with unregulated RTK activity.

## Methods

**TCPTP and variants**. Polymerase chain reaction (PCR) fragments of TCPTP (residues: 1–387) TCPTP$_{CAT}$ (catalytic domain, residues: 1–288, 1–302 or 1–314), TCPTP$_{Tail}$ (C-terminal tail, residues: 289–387 or 303–387) were amplified from full-length human TCPTP (TC45 isoform). TCPTP, TCPTP$_{CAT}$ (residues: 1–288, 1–314), and TCPTP$_{Tail}$ (residues: 289–387) fragments were cloned into pMCSG7 including a N-terminal His$_6$-tag using ligation-independent cloning[41]. For all NMR experiments, the PCR fragments of TCPTP, TCPTP$_{CAT}$ (residues: 1–302) and TCPTP$_{Tail}$ (residue: 303–387) were sub-cloned into pRP1B as previously described[42].

**Site-directed mutagenesis**. TCPTP$_{TailRK}$ (N373R/E374K/N375R/E376K), TCPTP-L1-loop (H30A/D31A/Y32A), and L6 loop (T106A/K107A) and (T106A/K107E) were generated using quick change site-directed mutagenesis protocol (Quikchange; Agilent). The TCPTP-L1-loop (H30A/D31A/Y32A/H34A/R35A) and TCPTP-C-terminal variants (R377A/K378A/R379A/K380A/R381A) were generated using the Q5-site directed mutagenesis protocol (New England BioLabs) using back-to-back primer design phosphorylated at the 5-prime end. All mutations were confirmed by sequencing.

**Protein expression and purification**. For protein expression, plasmid DNAs encoding TCPTP gene variants were transformed into *E. coli* BL21 (DE3) RIL cells (Agilent). Cells were grown in Luria Broth or M9 medium in the presence of selective antibiotics at 37 °C to an $OD_{600}$ of ~1.0 and expression was induced by the addition of 1 mM IPTG (Isopropyl β-D-1-thiogalactopyranoside). Induction proceeded for around 4 h at 37 °C (for TCPTP$_{Tail}$ constructs) or overnight at 18 °C (for TCPTP and TCPTP$_{CAT}$) prior to harvesting by centrifugation at 8,000 × g. Cell pellets were stored at −80 °C until purification.

For NMR measurements, expression of uniformly ($^1$H,$^{15}$N)-, ($^2$H,$^{15}$N)-, ($^1$H,$^{15}$N,$^{13}$C)-, or ($^2$H,$^{15}$N,$^{13}$C)- labeled proteins were facilitated by growing cells in $H_2O$ or $D_2O$ based M9 minimal medium containing 1 g/L $^{15}NH_4Cl$ and/or 4 g/L [$^1$H,$^{13}$C]- or [$^2$H,$^{13}$C]-D-glucose (CIL or Isotec) as the sole nitrogen and carbon sources, respectively. Multiple rounds (25, 50, 75, 90, and 100%) of $D_2O$ adaptation[43] were necessary for high-yield expression.

Cell pellets were lysed in Lysis buffer (50 mM Tris pH 8.0, 500 mM NaCl, 5 mM imidazole, 0.1% Triton X-100) containing EDTA-free protease inhibitor cocktail (Roche) using a high-pressure homogenizer (Avestin C3). The lysate was centrifuged at 42,000×g to separate soluble supernatant from insoluble cell pellets and filtered through a 0.22-μm PES filter before loading onto a His-trap HP column (GE Healthcare). After loading, the column was washed with Buffer A (50 mM Tris pH 8.0, 500 mM NaCl, 5 mM imidazole) and bound proteins were eluted by buffer B (50 mM Tris pH 8.0, 500 mM NaCl, 500 mM imidazole) using 5–500 mM imidazole gradient. Peak Fractions containing target protein were combined and dialyzed overnight at 4 °C in high salt Dialysis Buffer (50 mM Tris pH 8.0, 500 mM NaCl, 0.5 mM TCEP) with 5:1 volume ratio of TEV protease overnight. After dialysis cleaved His$_6$-tag and TEV were removed by reverse metal affinity chromatography (Subtraction). Flowthrough containing His$_6$-tag-cleaved protein were further purified by Size Exclusion Chromatography (SEC, Superdex-75, 26/60) in either storage buffer (50 mM Tris-HCL pH 7.4, 150 mM NaCl, 0.5 mM TCEP, 5% Glycerol) or NMR Buffer (20 mM HEPES pH 6.8, 150 mM NaCl, 0.5 mM TCEP). An additional heat purification step was applied to TCPTP$_{Tail}$ before SEC (80 °C for 10 min).

**NMR spectroscopy**. NMR data were collected on either Bruker NEO 600 MHz or 800 MHz spectrometers equipped with TCI HCN z-gradient cryoprobes at 298 K or 283 K. NMR measurements of TCPTP's variant construct were recorded using ($^1$H,$^{15}$N)-, ($^2$H,$^{15}$N)-, ($^1$H,$^{15}$N,$^{13}$C), or ($^2$H,$^{15}$N,$^{13}$C)-labeled protein at a final concentration of 0.6 mM for backbone assignment or 0.1 mM for titration in NMR buffer and 90% $H_2O$/10% $D_2O$. The sequence-specific backbone assignments of TCPTP (residues 1–387), TCPTP$_{CAT}$ (residues 1–302), TCPTP$_{Tail}$ (residues 303–387) and other variants were achieved using 3D triple resonance experiments including 2D [$^1$H,$^{15}$N] HSQC/TROSY, 3D HNCA, 3D HN(CO)CA, 3D HN(CO)CACB, and 3D HNCACB either as HSQC or TROSY variations. All NMR data were processed using TopSpin 4.0.5[44] and analyzed using either CARA (http://www.cara.nmr.ch) and/or CcpNMR[45].

**Chemical cross-linking**. TCPTP (0.5 μM) and complex mixture of TCPTP$_{CAT}$ (residue: 1–288) & TCPTP$_{Tail}$ (residue: 289–387) at 1:100 and 1:200 ratio (TCPTP$_{CAT}$ −0.75 μM while TCPTP$_{Tail}$ - 75 or 150 μM) were mixed with cross-linker BS3 (Bissulfosuccinimidyl suberate) or DSSO (Disuccinimidyl sulfoxide) in varying concentrations from 0 to 1000 μM in buffer (25 mM Hepes pH 7.4, 50 mM NaCl) and incubated at room temperature for 1 h. After incubation reaction was quenched by adding 1 M Tris Buffer (final concentration 50 mM). After stopping the reaction, cross-linked samples were subjected to SDS-PAGE followed by in-gel digestion prior to LC-MS or LC-MS$^n$ analysis.

Cross-linked TCPTP or TCPTP$_{CAT}$ + TCPTP$_{Tail}$ complex was excised from SDS-PAGE and reduced with 50 mM dithioerythritol (DTE) for 1 h at 37 °C, then alkylated with 100 mM iodoacetamide in the dark at room temperature. Following alkylation samples were sequentially digested by Lyc-C protease at 37 °C for 3 h, followed by trypsin at 37 °C overnight. After protease digestion, peptide mixtures were extracted and desalted on C18 Stage Tips using standardized protocols[46]. Eluted samples were dried using a SpeedVac vacuum concentrator and resuspended in 0.1% formic acid for further analysis.

**LC-MS analysis for BS3 cross-linked TCPTP**. Mass analysis was performed using an EASY-nLC$^{TM}$ 1200 system connected to a Thermo Orbitrap Fusion Lumos Tribrid Mass Spectrometer (ThermoFisher) equipped with a nano spray interface (New Objective). Peptide mixtures were loaded onto a 75-μm ID, 25 cm length PepMap C18 column (ThermoFisher) packed with 2 μm particles, pore size −100 Å and separated using a segmented gradient in 90 min from 5 to 45% solvent B (0.1% formic acid in acetonitrile) at a flow rate of 300 nl/min. Solvent A was 0.1% formic acid in water. The mass spectrometer was operated in the data-dependent mode. Briefly, survey scans of peptide precursors from 350 to 1600 *m/z* were performed at 120 K resolution with a $2 \times 10^5$ ion count target. Fragmentation (MS2) spectra were acquired in the Orbitrap at 60 K resolution. Precursor ions with 3–8 positive charges were selected for fragmentation with dynamic exclusion of 30 secs and isolation window of 1.6 Th. Additional MS2 settings included an automatic gain control target of 50,000 ions and a maximum injection time of

120 ms. The normalized collision energy was set to 30% for HCD scans. MS1 and MS2 scans were acquired in the Orbitrap.

**LC-MS$^n$ analysis for DSSO cross-linked sample**. Mass analysis was performed using an EASY-nLC$^{TM}$ 1200 system connected to a Thermo Orbitrap Fusion Lumos Tribrid Mass Spectrometer (Thermo Fisher) equipped with a nano spray interface (New Objective). Peptide mixtures were loaded onto a 75-μm ID, 25 cm length PepMap C18 column (Thermo Fisher) packed with 2 μm particles, pore size- 100 Å and separated using a segmented gradient in 90 min from 5 to 45% solvent B (0.1% formic acid in acetonitrile) at a flow rate of 300 nl/min. Solvent A was 0.1% formic acid in water. The mass spectrometer was operated in the data-dependent mode. Briefly, survey scans of peptide precursors from 350 to 1600 *m/z* were performed at 60 K resolution with a $2 \times 10^5$ ion count target. MS data acquisition methods used CID-MS2/EThcD-MS2/HCD-MS3 for DSSO cross-linked samples. Precursor ions with 3–8 positive charges were selected for CID-MS2/EThcD-MS2 fragmentation with dynamic exclusion of 30 secs and isolation window of 1.6 Th. Fragmentation (MS2) spectra were acquired in the Orbitrap at 30 K resolution. Subsequently, mass-difference-dependent HCD-MS3 acquisitions were triggered if a unique mass difference (Δ = 31.9721 Da) was observed in the CID-MS2 spectrum. The normalized collision energy was set to 25% for CID-MS2 scans and 30% for HCD-MS3 scans. MS1 and MS2 scans were acquired in the Orbitrap whereas MS3 scans were detected in the ion trap.

**Cross-linking data analysis and software parameters (PD-XLinkX)**. Raw MS files were analyzed using ThermoScientific$^{TM}$ Proteome Discoverer$^{TM}$ 2.4 software using the XLinkX node for cross-linked peptides and SEQUEST HT search engine for unmodified and dead-end modified peptides. Carbamidomethylation (+57.02146 Da) was used as a static modification for cysteine. Cross-linked mass modifications were used as variable modifications for lysine in addition to methionine oxidation (+15.99492 Da). For the search of inter-peptide cross-link by DSSO, cross-link modification was set as DSSO (+158.00376 Da), while for the mono-links peptide detection cross-link modification was set to DSSO-hydrolyzed (+176.01433 Da). For BS3 cross-link peptide search cross-link modification was set as BS3 (+138.06808 Da) to detect inter-peptide cross-link while for mono-links peptide cross-link modification was set to BS3-hydrolyzed (+156.07864 Da). The protease cleavage site was set to "Trypsin", and the maximum number of missed cleavage during proteolytic digestion was defined as 3. For the linear peptide search, precursor mass tolerance and fragment mass tolerance were set to 10 ppm and 0.02 Da respectively. For the cross-linked peptide search precursor mass tolerance, FTMS (Fourier transform mass spectrometry) fragment mass tolerance and ITMS (Ion trap mass spectrometry) fragment mass tolerance were set to 10 ppm, 20 ppm, and 0.5 Da respectively. Data were searched against TCPTP protein sequences (UniProt ID: P17706-2). FDR threshold was set to 0.05, and FDR strategy was defined as "Percolator". Cross-linked peptides were filtered to have a minimum threshold score of 20.

**SAXS data collection and processing**. Purified homogeneous solutions of TCPTP and TCPTP$_{CAT}$ (residue: 1–314) were characterized at the BL23A1 small- and wide-angle X-ray scattering (SWAXS) beamline at the National Synchrotron Radiation Research Center (NSRRC), Hsinchu, Taiwan. The inline HPLC-SAXS was controlled by an Agilent HPLC system. All samples were prepared within 24 h of data acquisition in SAXS buffer (50 mM Tris-HCl pH 7.4, 50 mM NaCl, 5% Glycerol and 1 mM TCEP). For the data collection, 100 μl of the sample (~15 mg/ml) was passed through either an Agilent Bio SEC-3, 300 Å, 4.6 × 300 mm (TCPTP$_{CAT}$) or GE Healthcare Superdex 200 Increase 5/150 GL column (TCPTP) at a flow rate of 0.035 ml/min. Consecutive frames of 20 s exposure time each at 15 °C was recorded according to the HPLC elution profile of each sample. SAXS data were collected with X-rays at a wavelength of λ = 0.8266 Å. The distance between the sample and detector was set at 2.5 meter. Normalization for beam intensity, buffer subtraction, and merging of the selected data were carried out using the NSRRC 23 A SWAXS data reduction package developed by NSRRC, Hsinchu Taiwan.

PRIMUS and GNOM (ATSAS software suite - 2.8.3) were used for initial SAXS data evaluation[47–49]. The data quality was assessed by inspection of the linearity of the Guinier region of the data. Structural parameter $R_g$, $I_{(0)}$, and $D_{max}$ were calculated using ScÅtter3. Porod Volume, Volume of correlation ($V_c$) and Molecular mass $M_r$ from ($V_c$) were determined using PRIMUS. A homology model of TCPTP was generated using Phyre2[50], which was used as an initial input model for conformational sampling in solution against our experimental SAXS data by using MultiFoXS server[32], which was also used to fit the averaged theoretical scattering intensity from the selected ensemble of conformations into the experimental SAXS data. The selected ensembles were automatically calculated from a pool of 10,000 independent models generated.

**Peptide synthesis**. All EGFR derived peptides used in activity assays (pEGFR peptide-1, from EGFR's cytoplasmic tail: $_{1059}$DDTFLPVPEpYINQSVPKR$_{1076}$, pEGFR peptide-2, from EGFR's kinase domain activation loop: $_{838}$LGAEEKEpY HAEGGKV$_{852}$), ITGA1 cytoplasmic tail (ITGA1_FCT: KIGFFKRPLKKKMEK, ITGA1_TCT: RPLKKKMEK, & ITGA1_TR: RPLKKKMEKRPLKKKMEK),

ITGA10 (KLGFFAHKKIPEEEKREEKLEQ), and ITGA11 (KLGFFRSARRRREPGL DPTPKVLE) were synthesized by the peptide synthesis core facility laboratory of the Institute of biological chemistry, Academia Sinica, Taiwan. After synthesis, the final purity of the peptides was determined to be ≥90% based on HPLC and MS analysis.

**EC$_{50}$ assays**. To determine the EC$_{50}$, synthesized ITGA1 peptides (ITGA1_FCT, ITGA1_TCT and ITGA1_TR) were mixed with TCPTP, and phosphatase activity was measured using the phospho-EGFR peptides (pEGFR peptide-1) as substrate. Lyophilized powder of all peptides was dissolved in 25 mM Hepes pH 7.4, 50 mM NaCl. For the assay, the TCPTP concentration was fixed at 1 nM and substrate phospho-peptide at 50 μM. The concentration of ITGA1 peptides was varied (ITGA1_FCT = 0 nM to 25600 nM, ITGA1_TCT = 0 μM to 512 μM & ITGA1_TR = 0 nM to 1024 nM) in a 200 μl reaction volume in assay buffer (25 mM Hepes pH 7.4, 50 mM NaCl, 1 mM DTT). The reaction is carried out at 30 °C for 30 min and stopped by adding 30 μl of phosphate reagent (phosphatase assay kit; Bio Vision) following the manufacturer's instruction. The absorbance of the reaction mixture was measured at 650 nm, which was converted to the amount of phosphate produced in reaction based on the standard curve generated from the known phosphate concentration. Data obtained from the reaction was fitted using nonlinear regression mode and EC$_{50}$ values were calculated based on the equation function "Log(agonist) vs. response-variable slope" using Graph Pad Prism 9.0.

**Phosphatase assay and kinetic analysis**. The rate of dephosphorylation of TCPTP and TCPTP$_{CAT}$ (residue 1–314) against 2 phosphotyrosine peptides derived from EGFR was monitored either with or without ITGA1 peptide (ITGA1_FCT & ITGA1_TR) in assay buffer (25 mM Hepes pH 7.4, 50 mM NaCl and 1 mM DTT). Lyophilized powder of all synthetic peptides was dissolved in assay buffer without DTT (phosphopeptides, 5 mM; ITGA1 peptides, 2 mM). For the kinetic analysis, the concentration of the enzyme was fixed at 1 nM while the substrate (phosphopeptides) concentration was varied from 2.5 μM to 160 μM in a total reaction volume of 200 μl. Effect of ITGA1 peptide on TCPTP & TCPTP$_{CAT}$ kinetics were assessed in saturated condition by adding an excess amount of ITGA1 peptide to the reaction mixture (ITGA1_TR = 1 μM; ITGA1_FCT = 25 μM). The reaction was incubated at 30 °C for 30 min and stopped by the addition of 30 μl phosphate reagent (phosphate colorimetric assay kit; Bio Vision) following the manufacturer's instructions. Data were normalized and fitted using a nonlinear curve, and kinetics parameters ($K_M$ and $k_{cat}$) were calculated from the Michaelis-Menten equation using Graph Pad Prism 9.0. All data points in the enzyme kinetics experiment were measured in triplicate, and the experiments were repeated three times.

Relative activity of TCPTP and its variant mutant construct were measured by adding 1 nM of enzymes against the fixed concentration of substrate (100 μM of phosphopeptides), the reaction was incubated at 30 °C for 30 min in assay buffer before getting stopped as described above. The amount of phosphate produced in each reaction at the end of incubation time was derived based on standard curve generated from known phosphate concentration, which was used for comparison of the relative activity of different construct. Statistical analysis (One-Way ANOVA with Tukey's multiple comparison test) was performed using Graph Pad Prism 9.0.

**DiFMUP assay**. Activity of TCPTP, TCPTP$_{CAT}$ (residue 1–314) either with or without ITGA1 peptides (ITGA1_TR; 1:50 molar ratio, ITGA1_FCT; 1:1000 molar ratio), PTP1B (residue 1–400) and PTP1B$_{CAT}$ (residue 1–321) were assessed in assay buffer (25 mM Hepes pH 7.4, 50 mM NaCl, and 1 mM DTT). The assay was performed in black polystyrene flat bottom 96-well plate. The concentration of DiFMUP was fixed at 10 μM and protein concentration at 2.5 nM in total reaction volume of 100 μl. The emitted fluorescent signal at 455 nm was continuously recorded for 25 min using a Tecan Infinite M1000 pro microplate reader (Tecan i-control, 2.0.10.0).

**TCPTP autoinhibition model**. We developed a TCPTP autoinhibition model to illustrate the interaction between the intrinsically disordered TCPTP C-terminal tail and its catalytic domain that integrates our biophysical and molecular results. Helix α7 (TCPTP residues 282–294) was added to the crystal structure of the TCPTP catalytic domain[51] (PDB ID: 1L8K) by superimposing PTP1B and copying the PTP1B α7 coordinates[28] (PDB ID: 5K9W), all structural additions/modulations were performed using the program Coot. Next, TCPTP residues 300–343, which we showed by NMR spectroscopy are intrinsically disordered and do not interact with the catalytic domain (no NMR intensity changes), were allowed to adopt multiple conformations (5 shown for simplicity), surrounding the TCPTP catalytic site. Our NMR experiment's (Fig. 3d, e) show that TCPTP residues 344–385 interact with TCPTP$_{CAT}$ at two surfaces: the N- and B-surfaces. TCPTP$_{Tail}$ residues 343–354 are modeled as a helix (residues 343–354 form the partially populated helix α8′ based on Cα/Cβ chemical shift analysis) to connect with the N-surface (TCPTP loops L1/2). TCPTP$_{Tail}$ residues 354–385 are modeled to wrap around TCPTP and interact with the B-surface (loops L6 and L14). Lastly, TCPTP residues E148 and D149 in β-strand β8 also exhibit CSPs, further positioning these residues within the proposed model and indicating that the C-terminal residues of the

TCPTP$_{Tail}$ likely extend to the front side and interact with E148 and D149 in β-strand β8.

**Reporting summary**. Further information on research design is available in the Nature Research Reporting Summary linked to this article.

## Data availability

All NMR chemical shifts have been deposited in the BioMagResBank with entry ID BMRB 50903, 50904, and 50905. Cross-linking mass spectrometry data have been deposited to the ProteomeXchange Consortium via the PRIDE[52] partner repository with the dataset identifier PXD029256. Two crystal structures published previously (PDB ID: 1L8K and PDB ID: 5K9W) were used in this study. Uncropped and unprocessed scans of the blots are provided in the Source Data file. Primers and recombinant DNA used in this study are summarized in Supplementary Table 5. The source data underlying Figs. 1d, 2a, b, 4a, b, 5a, 6a, d, Table-1, Supplementary Figs. 2a–c, 3a, b, 5a–c, 7b, c, 8b, and Supplementary Tables 1–4 are provided as a Source Data file. Source data are provided with this paper.

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

## Acknowledgements

We thank Y.-L. Hwang of the Synthesis Core Facility of Institute of Biological Chemistry, Academia Sinica, for peptide synthesis. We also thank the Biophysics Core Facility of Institute of Biological Chemistry, Academia Sinica, for supporting the biophysical analysis. The LC-MS/MS data were acquired at the Academia Sinica Common Mass Spectrometry Facilities for Proteomics and Protein Modification Analysis located at the Institute of Biological Chemistry, Academia Sinica, supported by Academia Sinica Core Facility and Innovative Instrument Project (AS-CFII-108-107). We thank the Academia Sinica High-Field NMR Center (HFNMRC) for supporting the initial NMR analyses. HFNMRC is funded by the Academia Sinica Core Facility and Innovative Instrument Project (AS-CFII-108-112). SAXS data were collected at the SAXS beamline, BL23A, Taiwan Light Source, National Synchrotron Radiation Research Center, Taiwan. This work is supported by the Taiwan Protein Project (Grant AS-KPQ-105-TPP) and the Academia Sinica Next-generation Pathway of Taiwan Cancer Precision Medicine Program (Grant AS-KPQ-107-TCPMP) to TCM, the American Diabetes Association Pathway to Stop Diabetes Grant 1-14-ACN-31 and grant R01NS091336 from the National Institute of Neurological Disorders and Stroke to WP, grant R01GM098482 from the National Institute of General Medicine to RP.

## Author contributions

J.P.S., Y.L., R.P., W.P., and T.C.M. developed the concept. J.P.S. designed, optimized and performed dephosphorylation, C.X.-M.S. and SAXS experiments with the help of Y.Y.C. and input from S.T.D.H.; Y.L. performed N.M.R. spectroscopy experiments with help of J.P.S., J.P.S., Y.L., R.P., W.P., and T.C.M. wrote the manuscript with comments and inputs from all co-authors.

## Competing interests

The authors declare no competing interests.
