## [Peer Review File · Nature Communications]

REVIEWER COMMENTS

Reviewer #1 (Remarks to the Author):

TCPTP is a ubiquitously expressed protein tyrosine phosphatase that generally displays low catalytic activity through a previously undescribed mechanism. Here, the authors demonstrate that an intrinsically disordered C-terminal tail of TCPTP inhibits access to the catalytic site without direct (high affinity) binding and that this inhibitory interaction is relieved by binding to known activators, here modeled using three integrin tails. This mechanism bears strong similarity to their recent demonstration of the autoinhibition mechanism in calcineurin, suggesting the generality of their “windshield wiper” model to phosphatase inhibition. The potential for similar mechanisms to manifest more broadly enhances the interest in this study for a broader readership.

This is a thoughtful study well described, with clear statistical analysis indicated in the figure legends and appropriate data deposition/availability. The minor critiques below are offered to help with the few sections of the manuscript that were less clear.

1. Based on Figure 1C, it is questionable whether full-length TCPTP alpha-nine-prime is assignable as a helix, while residues 315-340 are not. The magnitude of the SSP is similar. In contrast, it is clear that the assignment is justified by the data for the TCPTP-tail construct, suggesting that that these data facilitated the assignment in full-length as well. A bit of clarification in the text is probably merited.

2. In the discussion (see Page 13, Line 310) the authors claim to “show that the intrinsically low cellular activity of TCPTP is due to autoinhibition by its dynamic C-terminal IDR tail.” As there are no cellular assays in the reported work, this is an overstatement. The model supported by the study is likely to also be supported by future cell-based work and there is no reason to think that the tail conformational ensemble will change dramatically in the cellular environment. At least part of the cellular effect probably does come from the structural features identified here. However, it remains possible that other indirect effects or protein-protein interactions with as yet unidentified third partners contribute to the autoinhibition mechanism in the cell.

3. Page 3, Line 56 states “TCPTP (hereafter TC45),” which read as if to say that throughout the remainder of the manuscript the name TC45 would be used. Upon further reading, it seems the sentence meant that hereafter, the name TCPTP will exclusively refer to the TC45 isoform. A small rewording would help.

4. Page 4, Line 84: Minor grammatical problem with “via its cytoplasmic tail of ITGA1.” Should “its” be replaced with “the”?

5. Page 11, Line 260 “also recently to occur” should probably be “also recently seen to occur.”

6. Page 12 Line 283 the authors state that ITGA1 and the TCPTP tail bind the core domain “identically” based on similar chemical shift perturbations. This is probably semantic but stating that binding in two “fuzzy” complexes is “identical” seems a bit too strong.

7. Page 15, Line 368: again, this is probably just semantic, but it seems odd to state that the data “unravel” a new regulatory mechanism. An affirmative verb such as “reveal” may be better; “unravel” implies deconstruction of the accepted mechanism.

Reviewer #2 (Remarks to the Author):

In the manuscript “The catalytic activity of TCPTP is auto-regulated by its intrinsically disordered tail and activated by Integrin alpha-1,” Singh et al perform a series of biochemical and biophysical experiments to elucidate the mechanism of autoinhibition by the C-terminal tail of the protein tyrosine phosphatase TCPTP. Additionally, the authors provide evidence that activating peptides derived from collagen-binding integrin $\alpha 1$ (ITGA1) act by competing for the newly mapped C-terminal binding site on TCPTP’s catalytic domain, thereby releasing the auto inhibition caused by the C-terminal tail. The authors use an array of techniques including NMR, small-angle x-ray scattering, PTP activity assays, and chemical crosslinking/mass spectrometry that together provide a coherent picture of autoinhibition that is consistent with the authors’ central hypothesis.

The manuscript’s experiments are well designed and executed and expertly presented. Collectively, the paper’s biochemical and biophysical data provide the literature’s first detailed look at the structural basis for TCPTP autoinhibition. The work therefore constitutes a substantial contribution to the literature of PTP regulation, and I am happy to recommend the manuscript for publication in Nature Communications once the minor issues noted below are resolved.

1. What is the difference between Table 1 in the main manuscript and Table S1 in the supplemental information? The two tables appear to redundantly convey the same information, with Figure 1 just representing data from more replicates. Why include Table S1?
2. There are quite a few peptides used in the paper, so labeling axes as "Peptide (μM)," as in Fig. 2A, S3A, S6B, can be confusing. The axis labels should specify the peptide substrate being used.
3. Figure 2C: The PTP1B/TCPTP sequence alignment takes up a lot of figure space but doesn't provide much explanatory value, since TCPTP's mechanism of C-terminal autoinhibition is not shared by PTP1B. This figure could be moved to the supplementary information.
4. Figure 5A: Why are Michaelis-Menten kinetics parameters (k_{cat} and K_{m}) not reported for TCPTPL1 and TCPTPC-term? The authors should measure and report these parameters for completeness and consistency with the kinetic data on other TCPTP constructs.
5. Figure S6C: Why is there no TCPTP+ITGA1_FCT data (i.e., no green line) in Figure S6C? The authors should measure and report this data since the FCT peptide has the sequence of the full cytoplasmic tail of ITGA1 (as well as for completeness and consistency with the data in S6B).

Reviewer #3 (Remarks to the Author):

TCPTP targets a range of substrates including key kinases in cell growth/cancer and the immune response. Thus, the protein and general mechanism is a potential therapeutic target and an interesting yet challenging topic. The requirement for the TCPTP-tail region in TCPTP autoinhibition and the role of Integrin alpha-1 in activating TCPTP is known yet lacks a structural/mechanistic level of understanding.

Here, the authors use a combination of techniques, including NMR, cross-linking mass spectroscopy, and activity assays, to address the mechanism of TCPTP autoinhibition and release/activation by integrin alpha. Overall, this manuscript proposes an IDR-linked mechanism for TCPTP autoinhibition and competitive release by integrin alpha. However, despite a significant amount of work, the

experiments reported herein don't provide the mechanistic advances (or clarity) required to support the proposed models.

Major points

Page 8, Lines 186-188

"we added unlabeled TCPTP(TailRK) ... Small, but clear chemical shift perturbations (CSPs) were readily identified."

Because the NMR titration experiment (Figure S4D) indicates that the wild-type intrinsically disordered tail of TCPTP showed no significant perturbation of the resonances of the catalytic domain of TCPTP, it is unclear to what extent the construct TCPTP(TailRK), which harbors additional positively charged residues (373NENE376 -> 373RKRK376), would be a bona fide mimic in probing the binding site(s) on the catalytic domain.

As this point is critical for mapping binding details on the catalytic domain, it would be important to provide additional and strong evidence justifying the use of this construct in probing the binding sites. For example, comparison of CX-MS profiles of the catalytic domain in the presence of the wild-type tail (Figure S5) or the TailRK would lend support for their similarity in binding the catalytic domain. Additional *in trans* activity (inhibition) assay with the wild-type or mutant tail would shed light on their similarity in the regulation of the activity of the catalytic domain.

Page 8, Lines 186-188

"The CSPs map to 4 loops in TCPTP, which define two distinct surfaces: (1) the N-surface, defined by loops L1 and L2, ... and (2) the back or B-surface, defined by loops L6 and L14..."

It would be helpful to add the sequence information (for example, also in Figure 3) of the four loops so the reader can discern binding modes. Specifically, they would help identify (negatively charged) residues of the catalytic domain that potentially contact the positively charged segments of the tail. Furthermore, they would facilitate validation of binding mode(s) by experiments such as mutagenesis. Are negative charges present that support the proposed binding mode? Are these conserved? Can switch charge mutation be analyzed?

Page 10, Lines 240-242

"To test the importance of L1 for TCPTP C-terminus binding and autoinhibition, ..."

The activity assay using L1 mutant (TCPTP(L1)) provides validation for one of the potential binding sites on the catalytic domain. Further detailed analysis of which of the five mutated residues are key for binding would help delineate the binding mode. Is it a charge effect? It's unclear from these mutants. In addition, the validation of the other loops, including those that constitute the B-surface, would be desirable for comprehensive characterization of the binding sites on the catalytic domain. More detailed mutagenesis is pivotal in this section to identify the proposed charge effect. This is crucial given the use of the (373NENE376 -> 373RK RK376) mimic to identify the CSPs on the catalytic domain.

Figure 5

It is not clear how the data presented in Fig 1-4 (identifying regions that interact) provide the experimental resolution to support the interactions/basis for the 'windshield wiper' model proposed in 5B. How was the modelling conducted? How are the positions of tail secondary structure anchored on the catalytic domain surface (orange? 344-385) to then allow for the ~300-340 region to interact as an ensemble in the windshield wiper model?

Minor points:

Page 2, Lines 35-37

"we discovered that the C-terminal intrinsically disordered tail of TCPTP functions as ..."

Since the reference 29 [The noncatalytic C-terminal segment ... regulates activity via an intramolecular mechanism] reported the regulation of TCPTP activity by its C-terminal tail through intramolecular interaction, it would be worth considering revising this claim, and the title of the manuscript as well, to reflect the structural aspect of this study in advancing the understanding of TCPTP structure, function, and regulation. The title should reflect the structural findings of the paper.

Page 12, Line 282

"... revealed significant CSPs in the same residues that exhibit shifts when TCPTP(Tail) titrations"

The titration data (CSP) for TCPTP(TailRK), not TCPTP(Tail), was plotted in Figure 6C.

Methods

It is unclear how many times the XL-MS was repeated (n number) and if each cross-link was identified more than once. These data should be clarified in the methods. Cross-links reported in

Table S2 appear to be between the TCPTPCAT domains and the tail region. Were intramolecular cross-links detected between lysine residues within the TCPTPCAT (e.g. regions of known structure) that could be used to ascertain a false discovery rate? E.g. 137-144 etc. Figures for XL-MS should also be improved using available methods (e.g. xiNET, Circos plots) annotated with sec structural elements and identified tail binding regions.

Thursday, October 21, 2021

We would like to thank the reviewers for their insightful comments. We appreciate the opportunity to address the questions and comments raised about our manuscript by the reviewers.

As you will see, we have revised the manuscript to answer all questions of the reviewers, which clearly improved the manuscript. We have also performed additional experiments to further clarify and strengthen the data. Thus, we feel that we have fully addressed all the points raised. We hope that you will find the article with the included changes suitable for publication in *Nature Communications*.

In the response to the comments/questions, we have done the following to make our responses easy to read and identify:

- included all the Reviewer comments in ***bold italics***.
- Our responses are listed immediately below each point in standard font.

Reviewer #1 (Remarks to the Author):

TCPTP is a ubiquitously expressed protein tyrosine phosphatase that generally displays low catalytic activity through a previously undescribed mechanism. Here, the authors demonstrate that an intrinsically disordered C-terminal tail of TCPTP inhibits access to the catalytic site without direct (high affinity) binding and that this inhibitory interaction is relieved by binding to known activators, here modeled using three integrin tails. This mechanism bears strong similarity to their recent demonstration of the autoinhibition mechanism in calcineurin, suggesting the generality of their “windshield wiper” model to phosphatase inhibition. The potential for similar mechanisms to manifest more broadly enhances the interest in this study for a broader readership.

We thank the reviser for her/his strong support.

This is a thoughtful study well described, with clear statistical analysis indicated in the figure legends and appropriate data deposition/availability. The minor critiques below are offered to help with the few sections of the manuscript that were less clear.

We thank the reviser for her/his strong support. As detailed below, we have answered all minor critiques as carefully as possible.

1. Based on Figure 1C, it is questionable whether full-length TCPTP alpha-nine-prime is assignable as a helix, while residues 315-340 are not. The magnitude of the SSP is similar. In contrast, it is clear that the assignment is justified by the data for the TCPTP-tail construct, suggesting that that these data facilitated the assignment in full-length as well. A bit of clarification in the text is probably merited.

The reviewer is correct. There is no statistical relevance, and this was an oversight by us. We have revised the manuscript accordingly, i.e., helix $\alpha 9'$ has been deleted and helix $\alpha 10'$ is now the 'new' helix $\alpha 9'$ in the text and all figures.

2. In the discussion (see Page 13, Line 310) the authors claim to “show that the intrinsically low cellular activity of TCPTP is due to autoinhibition by its dynamic C-terminal IDR tail.” As there are no cellular assays in the reported work, this is an overstatement. The model supported by the study is likely to also be supported by future cell-based work and there is no reason to think that the tail conformational ensemble will change dramatically in the cellular environment. At least part of the cellular effect probably does come from the structural features identified here. However, it remains possible that other indirect effects or protein-protein interactions with as yet unidentified third partners contribute to the autoinhibition mechanism in the cell.

We fully agree with the reviewer. While it is very likely that our model will be supported by future cellular assay results, we have not performed them. We deleted the word 'cellular' as suggested.

3. Page 3, Line 56 states “TCPTP (hereafter TC45),” which read as if to say that throughout the remainder of the manuscript the name TC45 would be used. Upon further reading, it seems the sentence meant that hereafter, the name TCPTP will exclusively refer to the TC45 isoform. A small rewording would help.

The reviewer is correct; this was sloppy. This has been updated so it is 100% clear.

'TCPTP (hereafter referring to the TC45 isoform throughout the manuscript)'

4. Page 4, Line 84: Minor grammatical problem with “via its cytoplasmic tail of ITGA1.” Should “its” be replaced with “the”?

Thank you; updated.

5. Page 11, Line 260 “also recently to occur” should probably be “also recently seen to occur.”

Thank you; updated.

6. Page 12 Line 283 the authors state that ITGA1 and the TCPTP tail bind the core domain “identically” based on similar chemical shift perturbations. This is probably semantic but stating that binding in two “fuzzy” complexes is “identical” seems a bit too strong.

We appreciate the input of the reviewer. It is true that there are likely no two identical fuzzy complexes – due to the dynamic nature of these complexes. We reworded this statement to ‘in a similar manner’.

7. Page 15, Line 368: again, this is probably just semantic, but it seems odd to state that the data “unravel” a new regulatory mechanism. An affirmative verb such as “reveal” may be better; “unravel” implies deconstruction of the accepted mechanism.

Thank you; updated.

Reviewer #2 (Remarks to the Author):

In the manuscript “The catalytic activity of TCPTP is auto-regulated by its intrinsically disordered tail and activated by Integrin alpha-1,” Singh et al perform a series of biochemical and biophysical experiments to elucidate the mechanism of autoinhibition by the C-terminal tail of the protein tyrosine phosphatase TCPTP. Additionally, the authors provide evidence that activating peptides derived from collagen-binding integrin $\alpha 1$ (ITGA1) act by competing for the newly mapped C-terminal binding site on TCPTP’s catalytic domain, thereby releasing the auto inhibition caused by the C-terminal tail. The authors use an array of techniques including NMR, small-angle x-ray scattering, PTP activity assays, and chemical crosslinking/mass spectrometry that together provide a coherent picture of autoinhibition that is consistent with the authors’ central hypothesis.

The manuscript’s experiments are well designed and executed and expertly presented. Collectively, the paper’s biochemical and biophysical data provide the literature’s first detailed look at the structural basis for TCPTP autoinhibition. The work therefore constitutes a substantial contribution to the literature of PTP regulation, and I am happy to recommend the manuscript for publication in Nature Communications once the minor issues noted below are resolved.

We thank the reviser for her/his strong support of our work.

1. What is the difference between Table 1 in the main manuscript and Table S1 in the supplemental information? The two tables appear to redundantly convey the same information, with Figure 1 just representing data from more replicates. Why include Table S1?

Table 1 reports on the data acquired for pEGFR peptide-1 (tyrosine phosphorylated EGFR cytosolic tail peptide). Table S1 reports on the data acquired for pEGFR peptide-2 (tyrosine phosphorylated EGFR activation loop peptide). Indeed, as expected, the data are basically identical. Thus, Table S1 reports on different data and thus further highlights the rigor used in our studies.

Please see also our answer to reviewer 2 – question 2 below; this additional change should now further clarify the peptides used for the readers. Thank you for highlighting where we can improve the clarity of our work.

2. There are quite a few peptides used in the paper, so labeling axes as “Peptide (μM),” as in Fig. 2A, S3A, S6B, can be confusing. The axis labels should specify the peptide substrate being used.

Thank you for highlighting this issue and the solution recommended. We have taken this suggestion and now explicitly describe all peptides used in all figures. The peptides used for these studies were (and still are) described in the methods section under the header 'Peptide Synthesis'. Particularly, we used two distinct EGFR receptor-derived peptides. For clarity, we have now named them pEGFR peptide-1 and pEGFR peptide-2. Further, as recommended, we now use these names each time they are described in the text or are part of a figure. To this end, as suggested, we labeled all x-axes with the specific peptide substrate name to ensure there is no confusion for the readers.

3. Figure 2C: The PTP1B/TCPTP sequence alignment takes up a lot of figure space but doesn't provide much explanatory value, since TCPTP's mechanism of C-terminal autoinhibition is not shared by PTP1B. This figure could be moved to the supplementary information.

We appreciate the point the reviewer is making, but we prefer to keep the figure in the main text because we believe it is critical for the central discoveries/messages of the manuscript. First, it clearly illustrates that the two proteins are 72% identical in the catalytic domain (as referred to by the reviewer). Second, it also illustrates that TCPTP and PTP1B are, conversely, completely dissimilar in the C-terminal tail IDR domain. Third, it illustrates that the TCPTP C-terminal IDR domain is highly enriched in charged residues. These data are readily identified in this figure and these data are essential to understand both the rationale and outcomes of the presented data/work. Finally, different parts of the sequence, such as the loops that are part of the N/B-surface, are highlighted as well (as requested by Reviewer 3).

4. Figure 5A: Why are Michaelis-Menten kinetics parameters (k_{cat} and K_m) not reported for TCPTPL1 and TCPTP_{C-term}? The authors should measure and report these parameters for completeness and consistency with the kinetic data on other TCPTP constructs.

As requested, we also now show the Michaelis-Menten kinetics parameters (k_{cat} and K_m) for the two variants (TCPTP_{C-term} and TCPTP_{L1}). The new results are included in the updated version of Table 1, which is attached below for evaluation by the reviewer (also updated in the manuscript).

Table 1:

	k_{cat} (s^{-1})	K_m (μM)	k_{cat}/K_m ($\mu M^{-1}s^{-1}$)	N
TCPTP				
TCPTP _{CAT}	34.0 ± 0.9	24.8 ± 2.0	1.4 ± 0.1	9
TCPTP _{C-term}	23.7 ± 0.5	17.9 ± 1.1	1.3 ± 0.1	12
TCPTP _{L1}	19.3 ± 0.8	21.0 ± 2.7	0.9 ± 0.1	12
TCPTP	12.2 ± 0.2	12.4 ± 0.6	1.0 ± 0.1	9
ITGA1 (saturated)				
TCPTP _{CAT} + ITGA1_TR	31.1 ± 0.7	23.0 ± 1.6	1.3 ± 0.1	9
TCPTP + ITGA1_FCT	26.6 ± 0.8	20.2 ± 1.9	1.3 ± 0.1	9
TCPTP + ITGA1_TR	26.2 ± 0.6	15.9 ± 1.3	1.6 ± 0.1	9

Please find this new data also below in a figure for easier comparison of the catalytic activity between TCPTP_{C-term} and TCPTP_{L1} (wt-TCPTP [blue] control) using the pEGFR peptide-1 as substrate (data are presented as mean ± SE, n=12). Kinetic parameters derived from these data are included in Table 1; thus, we do not show the figure in the revised manuscript but decided to show it below for effortless evaluation by the reviewer.

5. Figure S6C: Why is there no TCPTP+ITGA1_FCT data (i.e., no green line) in Figure S6C? The authors should measure and report this data since the FCT peptide has the sequence of the full cytoplasmic tail of ITGA1 (as well as for completeness and consistency with the data in S6B).

As requested, we have now added this data to the Figure S7C (number has changed as we included a new Figure S6). Thank you for pointing this out so we present fully complete and consistent data. The updated Figure S7C (below for evaluation by the reviewer) is included in the revised manuscript.

Reviewer #3 (Remarks to the Author):

TCPTP targets a range of substrates including key kinases in cell growth/cancer and the immune response. Thus, the protein and general mechanism is a potential therapeutic target and an interesting yet challenging topic. The requirement for the TCPTP-tail region in TCPTP autoinhibition and the role of Integrin alpha-1 in activating TCPTP is known yet lacks a structural/mechanistic level of understanding.

We appreciate the support of the reviewer – highlighting the biological importance of TCPTP and the fact that we lack a molecular understanding of the regulation of TCPTP, which we are providing with the presented data.

Here, the authors use a combination of techniques, including NMR, cross-linking mass spectrometry, and activity assays, to address the mechanism of TCPTP autoinhibition and release/activation by integrin alpha. Overall, this manuscript proposes an IDR-linked

mechanism for TCPTP autoinhibition and competitive release by integrin alpha. However, despite a significant amount of work, the experiments reported herein don't provide the mechanistic advances (or clarity) required to support the proposed models.

We are providing detailed responses to all points raised by the reviewer below and we are confident that our data fully support our proposed model.

Major points

Page 8, Lines 186-188

“we added unlabeled TCPTP(TailRK) ... Small, but clear chemical shift perturbations (CSPs) were readily identified.”

Because the NMR titration experiment (Figure S4D) indicates that the wild-type intrinsically disordered tail of TCPTP showed no significant perturbation of the resonances of the catalytic domain of TCPTP, it is unclear to what extent the construct TCPTP(TailRK), which harbors additional positively charged residues (373NENE376 -> 373RKRK376), would be a bona fide mimic in probing the binding site(s) on the catalytic domain.

As this point is critical for mapping binding details on the catalytic domain, it would be important to provide additional and strong evidence justifying the use of this construct in probing the binding sites. For example, comparison of CX-MS profiles of the catalytic domain in the presence of the wild-type tail (Figure S5) or the TailRK would lend support for their similarity in binding the catalytic domain. Additional in trans activity (inhibition) assay with the wild-type or mutant tail would shed light on their similarity in the regulation of the activity of the catalytic domain.

Figure S4 clearly shows an interaction between TCPTP_{CAT} and TCPTP_{Tail}. In particular, as seen in Figure S4B, the addition of TCPTP_{CAT} to ¹⁵N-labeled TCPTP_{Tail} results in significant changes (reductions) in peak intensities of a large number of TCPTP_{Tail} cross-peaks (2D [¹H, ¹⁵N] HSQC data is shown). Furthermore, in Figure S4C, we quantify the observed peak intensity reduction and compare it with the peak intensity data for the TCPTP_{Tail} alone. TCPTP_{Tail} residues 345-380 show a clear reduction of peak intensities. Many groups, including ours have shown, that weak charge:charge ‘fuzzy’ protein:protein interactions between an IDP/IDR and a folded protein commonly lead to similar intensity peak reductions; i.e. the outcome of this interaction study is exactly as expected for a ‘fuzzy’ protein interaction.

The reviewer is correct – the reverse titration does not show chemical shift perturbations (CSPs) in the 2D [¹H, ¹⁵N] TROSY spectrum of the folded TCPTP_{CAT} at a 1:20 ratio (as is often seen for weak charge:charge ‘fuzzy’ protein:protein interactions; these can be more readily detected using the IDP than the folded protein). Based on this data we increased the charge:charge potential by creating TCPTP_{TailRK}. We performed the sequence-specific backbone assignment of TCPTP_{TailRK} and showed that upon addition of TCPTP_{CAT} the 2D [¹H, ¹⁵N] HSQC spectrum of TCPTP_{TailRK} showed alike intensity changes (only stronger, per design) than the 2D [¹H, ¹⁵N] HSQC spectrum of TCPTP_{Tail} confirming the identical mode of interaction.

Increasing charge to better highlight fuzzy charge:charge interactions is an established strategy used by many groups. Indeed, as the reviewer highlights, we have also performed complimentary CX-MS measurements with wild-type proteins, which showed results identical to our NMR data. Lastly, we used variant studies to further confirm the data (data shown in Figures 5 and S6). Thus, by testing the interaction using rigorous, orthogonal experimental approaches, with identical outcomes, we are confident that our data is consistent and as carefully interpreted as possible, as also highlighted by the reviews of reviewer 1 and 2.

Page 8, Lines 186-188

“The CSPs map to 4 loops in TCPTP, which define two distinct surfaces: (1) the N-surface, defined by loops L1 and L2, ... and (2) the back or B-surface, defined by loops L6 and L14...”

It would be helpful to add the sequence information (for example, also in Figure 3) of the four loops so the reader can discern binding modes. Specifically, they would help identify (negatively charged) residues of the catalytic domain that potentially contact the positively charged segments of the tail. Furthermore, they would facilitate validation of binding mode(s) by experiments such as mutagenesis. Are negative charges present that support the proposed binding mode? Are these conserved? Can switch charge mutation be analyzed?

The protein sequence of TCPTP is shown in Figure 2C and the discussed loops are highlighted in this Figure as requested. In addition, all the identified residues of the catalytic domain involved in interaction with the tail are labeled and color coded in Figures 2C, 3D and 3E. As requested, the new results for switch charge mutation have been provided in the answer to Reviewer 3's next question.

Page 10, Lines 240-242

**“To test the importance of L1 for TCPTP C-terminus binding and autoinhibition, ...”
The activity assay using L1 mutant (TCPTP(L1)) provides validation for one of the potential binding sites on the catalytic domain. Further detailed analysis of which of the five mutated residues are key for binding would help delineate the binding mode. Is it a charge effect? It's unclear from these mutants. In addition, the validation of the other loops, including those that constitute the B-surface, would be desirable for comprehensive characterization of the binding sites on the catalytic domain. More detailed mutagenesis is pivotal in this section to identify the proposed charge effect. This is crucial given the use of the (373NENE376 -> 373RKRK376) mimic to identify the CSPs on the catalytic domain.**

As requested, we have performed further mutagenesis experiments to corroborate the essential role of charge:charge interactions for the recruitment of the TCPTP C-terminal tail. We focused on residues in TCPTP loops L1 and L6. A new Figure S6 (also shown below for evaluation by the reviewer) is included in the revised manuscript.

Two variants of loop L1 (L1 variant 1 & L1 variant 2) were generated:

- *Variant 1:* H30A/D31A/Y32A. Interaction of TCPTP_{TailRK} is basically identical between TCPTP_{CAT} and TCPTP_{CAT_L1_H30A/D31A/Y32A}.
- *Variant 2:* H30A/D31A/Y32A/H34A/R35A (TCPTP_{L1} variant described in the manuscript). Interaction of TCPTP_{TailRK} is slightly weakened between TCPTP_{CAT} and TCPTP_{CAT_L1_H30A/D31A/Y32A/H34A/R35A}.

The 2D [¹H, ¹⁵N] HSQC spectrum of TCPTP_{TailRK} with TCPTP_{CAT} L1 variant 1 showed a nearly identical reduction in cross-peak intensity as observed with wt-TCPTP_{CAT}, indicating that TCPTP residues 30/31/32 are not critical for the interaction. Repeating the experiment with TCPTP_{CAT} L1 variant 2 showed a slight increase in cross-peak intensity compared with either wt-TCPTP_{CAT} or TCPTP_{CAT} L1 variant 1, indicating weakened binding with TCPTP_{TailRK}. The results indicate that deletion of charged residues (H30A/D31A/Y32A/H34A/R35A) leads to a weakening of binding with TCPTP_{TailRK}, highlighting the importance of charge in L1.

Two loop L6 variants (L6 variant 1 & L6 variant 2) were generated:

Loop 6:

- *Variant 1*: T106A/K107A. Interaction of TCPTP_{TailRK} is basically identical between TCPTP_{CAT} and TCPTP_{CAT_L6_T106A/K107A}.
- *Variant 2*: T106A/K107E (charge reversal – increasing negative charge is expected to lead to a stronger interaction). Interaction of TCPTP_{TailRK} is stronger between TCPTP_{CAT} and TCPTP_{CAT_L6_T106A/K107E}.

The 2D [¹H,¹⁵N] HSQC spectrum of TCPTP_{TailRK} with TCPTP_{CAT} L6 variant 1 showed a nearly identical reduction in cross-peak intensity as observed with wt-TCPTP_{CAT}. On the other hand, TCPTP_{CAT} L6 variant 2 showed a decrease in cross-peak intensity compared with either wt-TCPTP_{CAT} or TCPTP_{CAT} L6 variant 1. The charge reversal of K107E in TCPTP_{CAT} L6 variant 2 leads to expected stronger binding with TCPTP_{TailRK}, highlighting the importance of charge in loop L6.

These data are fully consistent with our proposed ‘fuzzy’ charge:charge interaction.

Figure 5

It is not clear how the data presented in Fig 1-4 (identifying regions that interact) provide the experimental resolution to support the interactions/basis for the ‘windshield wiper’ model proposed in 5B. How was the modelling conducted? How are the positions of tail secondary structure anchored on the catalytic domain surface (orange? 344-385) to then allow for the ~300-340 region to interact as an ensemble in the windshield wiper model?

We thank the reviewer for this suggestion. We now provide additional details on how the model displayed in Figure 5B was created in the updated methods section. We feel that such models are helpful to better understand such complex interactions, while we agree with the reviewer that it is simply impossible to ‘catch’ all possible binding modes. Critically, this is the big difference between *an interaction between 2 folded proteins*, which can form a major stabilized interaction platform and *the interaction between an IDP and a folded protein*; i.e., these ‘fuzzy’ complexes are impossible to 100% correctly visualize. Here we provide the best possible model, which is consistent with all of our data. We are confident that such a model is helpful for most readers to better understand the data and allow them to better design their own experiments, based on our data presented.

Added text:

‘We developed a TCPTP auto-inhibition model to illustrate the interaction between the intrinsically disordered TCPTP C-terminal tail and its catalytic domain that integrates our biophysical and molecular results. Helix $\alpha 7$ (TCPTP residues 282-294) was added to the crystal structure of the TCPTP catalytic domain (PDB ID: 1L8K) by superimposing PTP1B and copying the PTP1B $\alpha 7$ coordinates (PDB ID: 5K9W)]; all structural additions/modulations were performed using the program Coot. Next, TCPTP residues 300-343, which we showed by NMR spectroscopy are intrinsically disordered and do not interact the catalytic domain (no NMR intensity changes), were allowed to adopt multiple conformations (5 shown for simplicity), surrounding the TCPTP catalytic site. Our NMR experiments (Figs. 3D, E) show that TCPTP residues 344-385 interact with TCPTP_{CAT} at two surfaces: the N- and B-surfaces. TCPTP_{Tail} residues 343-354 are modeled as a helix (residues 343-354 form the partially populated helix $\alpha 8'$ based on $C\alpha/C\beta$ chemical shift analysis) to connect with the N-surface (TCPTP loops L1/2). TCPTP_{Tail} residues 354-385 are modeled to wrap around TCPTP and interact with the B-surface (loops L6 and L14). Lastly, TCPTP residues E148 and D149 in β -strand $\beta 8$ also exhibit CSPs, further positioning these residues within the proposed model and indicating that the C-terminal residues of the TCPTP_{Tail} likely extend to the front side and interact with E148 and D149 in β -strand $\beta 8$.’

Minor points:

Page 2, Lines 35-37

“we discovered that the C-terminal intrinsically disordered tail of TCPTP functions as ...” Since the reference 29 [The noncatalytic C-terminal segment ... regulates activity via an intramolecular mechanism] reported the regulation of TCPTP activity by its C-terminal tail through intramolecular interaction, it would be worth considering revising this claim, and the title of the manuscript as well, to reflect the structural aspect of this study in advancing the understanding of TCPTP structure, function, and regulation. The title should reflect the structural findings of the paper.

As requested, we changed the wording; instead of 'discovered' we use 'show'. We prefer to keep the title as is.

Page 12, Line 282

"... revealed significant CSPs in the same residues that exhibit shifts when TCPTP(Tail) titrations"

The titration data (CSP) for TCPTP(TailRK), not TCPTP(Tail), was plotted in Figure 6C.

Updated as requested.

Methods

It is unclear how many times the XL-MS was repeated (n number) and if each cross-link was identified more than once. These data should be clarified in the methods. Cross-links reported in Table S2 appear to be between the TCPTPCAT domains and the tail region. Were intramolecular cross-links detected between lysine residues within the TCPTPCAT (e.g. regions of known structure) that could be used to ascertain a false discovery rate? E.g. 137-144 etc. Figures for XL-MS should also be improved using available methods (e.g. xiNET, Circos plots) annotated with sec structural elements and identified tail binding regions.

We thank the reviewer for the constructive comments regarding the methods and data interpretation of our cross-linking coupled with mass spectrometry (CX-MS) data. Each CX-MS dataset was derived from a single experiment. For each analyzed construct (either full-length TCPTP or TCPTP_{CAT}+TCPTP_{Tail}), we used two cross-linkers, DSSO and BS3, in independent experiments. The idea behind using two cross-linkers in independent experiments instead of repeating the experiment by the same cross-linker was to avoid the possibility of missing any interaction site due to the specific reactivity of cross-linker. By doing so, we ensured that the binding interfaces between the TCPTP C-terminal tail and the TCPTP catalytic domain were accurately and as complete as possible mapped. Our CX-MS data (cis, mapped in Figure 4 and detailed in Table S2; trans, mapped in Figure S5 and detailed in Table S3) showed basically identical binding sites, consistent with the NMR spectroscopy results. Within each experimental setup, the number of cross-linked spectrum matches (CSMs) for any identified peptide pair is shown in the resource data file. It should be noted that for most of the cross-linked peptide pairs, more than one CSM was found. Furthermore, we have manually analyzed each cross-linked peptide pairs shown in Tables S2 and S3 to ensure data accuracy. According to our data processing criteria described in method section of the manuscript, we did not observe intramolecular cross-link between lysine residues within the TCPTP_{CAT} domain.

We appreciate the suggestion of annotating the CX-MS results by available methods such as xiNET and Circos plots. In fact, xiNET format was adopted by us for generating the crossed linked map shown in Figure 4B and Figure S4B (simplified version for easy understanding by the readers). In order to provide an additional presentation of the CX-MS data, we have now included the plots annotated by CX-Circos in Figure S5C.

All requested changes have been performed and the manuscript has been significantly revised.

In closing, we would like to thank the reviewers for their careful and insightful comments. After responding to the comments of the reviewers and updating the manuscript accordingly, we hope that you will find this manuscript acceptable as an Article for *Nature Communications*.

Sincerely,

Wolfgang Peti & Tzu-Ching Meng
Corresponding authors

REVIEWERS' COMMENTS

Reviewer #3 (Remarks to the Author):

A clear and detailed response is provided that covers all initial concerns. This includes the addition of new supplemental data (mutants) and further details on methodology. Happy to recommend the manuscript for publication.